# Skeletal muscle enhancer interactions identify genes controlling whole-body metabolism

Kristine Williams[1], Lars R. Ingerslev [1], Jette Bork-Jensen[1], Martin Wohlwend[2], Ann Normann Hansen[1], Lewin Small [1], Rasmus Ribel-Madsen[1], Arne Astrup [3], Oluf Pedersen[1], Johan Auwerx [2], Christopher T. Workman [4], Niels Grarup [1], Torben Hansen [1] & Romain Barrès [1 ✉]

Obesity and type 2 diabetes (T2D) are metabolic disorders influenced by lifestyle and genetic factors that are characterized by insulin resistance in skeletal muscle, a prominent site of glucose disposal. Numerous genetic variants have been associated with obesity and T2D, of which the majority are located in non-coding DNA regions. This suggests that most variants mediate their effect by altering the activity of gene-regulatory elements, including enhancers. Here, we map skeletal muscle genomic enhancer elements that are dynamically regulated after exposure to the free fatty acid palmitate or the inflammatory cytokine TNFα. By overlapping enhancer positions with the location of disease-associated genetic variants, and resolving long-range chromatin interactions between enhancers and gene promoters, we identify target genes involved in metabolic dysfunction in skeletal muscle. The majority of these genes also associate with altered whole-body metabolic phenotypes in the murine BXD genetic reference population. Thus, our combined genomic investigations identified genes that are involved in skeletal muscle metabolism.

[1] Novo Nordisk Foundation Center for Basic Metabolic Research, Faculty of Health and Medical Sciences, University of Copenhagen, Copenhagen, Denmark. [2] Laboratory of Integrative and Systems Physiology, Interfaculty Institute of Bioengineering, École Polytechnique Fédérale de Lausanne, Lausanne, Switzerland. [3] Department of Nutrition, Exercise and Sports Science, Faculty of Science, University of Copenhagen, Copenhagen, Denmark. [4] Department of Biotechnology and Biomedicine, Technical University of Denmark, Kongens Lyngby, Denmark. ✉email: barres@sund.ku.dk

The prevalence of obesity and T2D comorbidity is reaching epidemic proportions worldwide, with currently 1.9 billion adults estimated as being overweight or obese[1] and 380 million suffering from T2D[2]. Skeletal muscle constitutes the largest metabolic organ and accounts for 30% of the basal metabolic rate[3], and as the most prominent site of insulin-mediated glucose uptake in humans, insulin resistance (IR) in muscle is considered a contributing defect during development of T2D[4]. While the molecular basis for the pathology of obesity and T2D is incompletely understood, it is clear that both genetic and environmental factors contribute, probably in a synergistic manner[5]. Genome-wide association studies (GWAS) have identified a plethora of genetic variants associated with T2D and obesity traits[6–8]. However, only a minority (<5%) of GWAS identified variants are located in coding sequences[9], which makes functional characterization complex. Several studies have identified that a substantial proportion of the disease-associated variants lie within regulatory regions, including enhancer elements[9–11].

Enhancers serve as binding sites for transcription factors and co-regulators that assist in DNA looping and recruitment of the transcriptional machinery to targeted promoters. With an estimated 50,000 to 100,000 active enhancers in any given mammalian cell type[12], enhancers are thought to account for the complexity of gene regulation. Enhancers are characterized by the presence of histone modifications including monomethylation of histone 3 lysine 4 (H3K4me1) and acetylation of histone 3 lysine 27 (H3K27ac)[13–15]. Thus, by determining the genome-wide distribution of these histone marks, it is possible to generate genome-wide maps of active enhancers (the enhancerome) in a specific tissue. Mapping the enhancerome in various cell types and during embryonic stem cell differentiation has demonstrated that enhancer activation is highly cell-type specific and dynamic[16,17], and several studies have proposed that impaired enhancer activation could be at the origin of disease[18–21]. Besides interacting with nearby promoters, enhancers also engage in long-range interactions. Indeed, it is estimated that approximately 35–40% of all promoter-enhancer interactions are intervened by at least one gene[22], which makes exact enhancer-target prediction challenging. Long-range enhancers interactions can be identified by chromosome conformation capture methods[23,24].

In the present study, we aimed to identify target genes of GWAS SNPs in human skeletal muscle by using cultured myotubes subjected to metabolic stress by either palmitate or TNFα exposure. Elevation of plasma levels of free fatty acids and proinflammatory cytokines associates with increasing adiposity[25] and represent an important link between obesity, skeletal muscle IR, and T2D. By RNA profiling and genome-wide mapping of enhancer elements in myotubes, we found that palmitate or TNFα treatment led to massive changes in gene transcription, as well as alterations in the activity of enhancers. Moreover, we showed that enhancers regulated by palmitate or TNFα exposure, overlapped SNPs from GWAS of BMI, waist-to-hip ratio (WHR), IR or T2D. Moreover, by mapping global promoter-enhancer interactions by chromatin conformation analysis, we directly couple these enhancers to promoters, where we found a concurrent change in gene transcription by the respective treatments. Thus, we established physical links between numerous GWAS SNPs and muscle-expressed genes and provided insight into the association between the identified genes and metabolic function in vivo.

## Results

**Transcriptomic profiling of human skeletal muscle cells**. To study concurrent changes in gene transcription, enhancer activities and chromatin conformation, we used primary human skeletal muscle cells differentiated into myotubes that were subjected to metabolic stress by treatment with either palmitate or TNFα (Supplementary Fig. 1A). As previously reported[26–29], both treatments lowered insulin sensitivity, as confirmed by decreased AKT Ser-473 phosphorylation in response to insulin stimulation (Supplementary Fig. 1B–E).

First, we performed transcriptomic analysis by RNA-sequencing (RNA-seq). Multidimensional Scaling (MDS) plots showed a clear sample separation based on palmitate or TNFα treatment (Fig. 1a). In total, we detected expression of 14,402 genes in skeletal muscle cells, of which 1542 were regulated by palmitate treatment (621 downregulated and 921 upregulated; Fig. 1b, Supplementary Data 1, and Supplementary Fig. 2A) and 4522 were changed by TNFα treatment (2247 downregulated and 2275 upregulated; Fig. 1c, Supplementary Data 1, and Supplementary Fig. 2B). Gene ontology (GO) analysis of the differentially expressed genes (Supplementary Data 2) demonstrated strong upregulation of genes involved in lipid metabolism, as well as regulation of inflammatory responses (Fig. 1d) by palmitate exposure, whereas terms related to nucleosome assembly were specifically downregulated (Fig. 1e). GO analysis of TNFα upregulated genes returned several terms related to immune signaling (Fig. 1f), whereas downregulated genes were related to protein targeting to the endoplasmic reticulum (ER), insulin-like growth factor signaling and muscle filament sliding (Fig. 1g). Interestingly, both treatments seemed to significantly upregulate genes involved in inflammation (Supplementary Data 2 and Fig. 1h), and to downregulate genes related to muscle contraction (Supplementary Data 2 and Fig. 1i), both of which are processes related to skeletal muscle dysfunction and insulin resistance. Thus, our transcriptomic analyses of human muscle myotubes reveal thousands of target genes of which many are related to metabolic dysfunction.

**The dynamic enhancerome of skeletal muscle cells**. Through chromatin immunoprecipitation followed by sequencing (ChIP-seq), we mapped the distribution of the enhancer-associated histone H3 modifications, H3K4me1 and H3K27ac, in the muscle myotubes treated with TNFα or palmitate. Genome-wide, we identified 107,405 and 80,388 significant peaks of H3K4me1 or H3K27ac, respectively (Fig. 2a). These were mostly located in non-coding DNA, such as introns and intergenic regions, as well as in promoters (Supplementary Fig. 3). In order to identify enhancers, we subtracted active promoter regions (defined by the promoter-associated H3K4me3 mark). We found that most (95.5%) of the non-promoter associated H3K27ac peaks overlapped a H3K4me1 peak, whereas only 36.9% of the H3K4me1 peaks overlapped H3K27ac (Fig. 2a). These findings support the notion that enhancers can be primed (marked by only H3K4me1) or active (marked by both H3K4me1 and H3K27ac)[16,17]. MDS plots of non-promoter associated H3K27ac and H3K4me1 ChIP-seq data demonstrated a clear treatment-based separation of samples for H3K27ac (Fig. 2b), whereas this was less obvious for H3K4me1 (Fig. 2c), underlining the assumption that especially H3K27ac undergoes dynamic regulation in response to external stimuli and determines enhancer activity[16,30,31]. Therefore, to identify enhancers that were differentially activated after palmitate or TNFα treatment, we searched for peaks within the 62,866 identified active enhancers (covered by both H3K4me1 and H3K27ac) that showed significant changes in H3K27ac levels. This analysis returned 2243 enhancers with altered activity after palmitate treatment (FDR < 0.01, 1190 with a decreased activity and 1053 with an increased activity) (Fig. 2d and Supplementary Data 3), and 17,037 enhancers that changed activity after TNFα treatment (FDR < 0.01, 12,380 with a decreased activity and 4,657

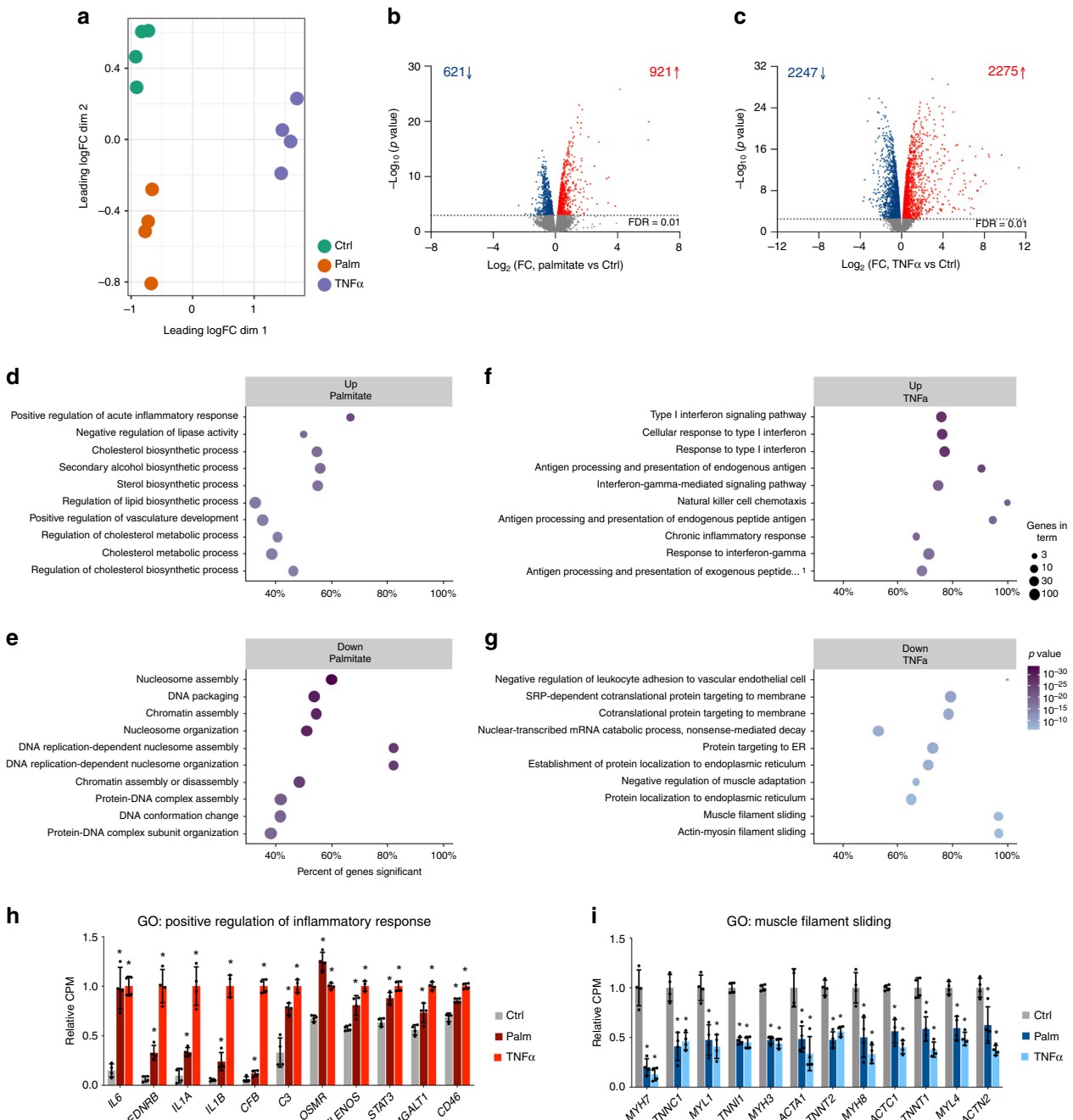

**Fig. 1 Gene expression analysis after palmitate or TNFα treatment. a** MDS plot of RNA-seq data from control (ctrl), palmitate (palm) or TNFα treated human skeletal myotubes. Leading log fold-change (logFC) is the mean logFC between the 500 most divergent genes between each pair of samples. On the axes, "dim" means dimension. **b**, **c** Volcano plot representation of genes regulated by palmitate (**b**) or TNFα (**c**). Blue dots represent genes that are significantly downregulated and red dots represent genes that are upregulated by the respective treatments (n = 4 biological replicates, FDR < 0.01). **d**, **e** Top 10 GO terms upregulated (**d**) or downregulated (**e**) by palmitate. **f**, **g** Top 10 GO terms upregulated (**f**) or downregulated (**g**) by TNFα. The x-axis shows the percent of genes in the category that are differentially expressed with an FDR < 0.01. The legend shows 'Genes in term', which is the number of genes expressed in these samples. The P-value is calculated using the CAMERA method. All terms have an FDR of less than 0.0001. **h** Examples of palmitate and TNFα upregulated genes related to acute inflammatory response. Relative CPM indicates RNA-seq counts per million relative to TNFα treatment. Values are represented as the mean ± S.D. (n = 4 biological replicates, *FDR < 0.01). **i** Examples of palmitate and TNFα downregulated genes related to muscle filament sliding. Relative CPM indicates RNA-seq counts per million relative to control. Values are represented as the mean ± S.D. (n = 4 biological replicates). Asterisks indicate genes that are significantly regulated in the RNA-seq analysis (*FDR < 0.01).

with an increased activity) (Fig. 2e and Supplementary Data 3). Examples of enhancers with a strong increase in H3K27ac after palmitate treatment included elements located 10 kb upstream of the *PDK4* promoter and 9 kb upstream of *ANGPTL4* (Fig. 2f, g)—two genes known to play a role in fatty acid metabolism.

Moreover, some enhancers strongly regulated by TNFα were located close to cytokine genes, exemplified by enhancers located 21 kb downstream of *CCL11* and 17 kb upstream of *CCXL8* (Fig. 2h, i). The changes in H3K27ac were validated independently by ChIP-qPCR (Supplementary Fig. 4A), which further

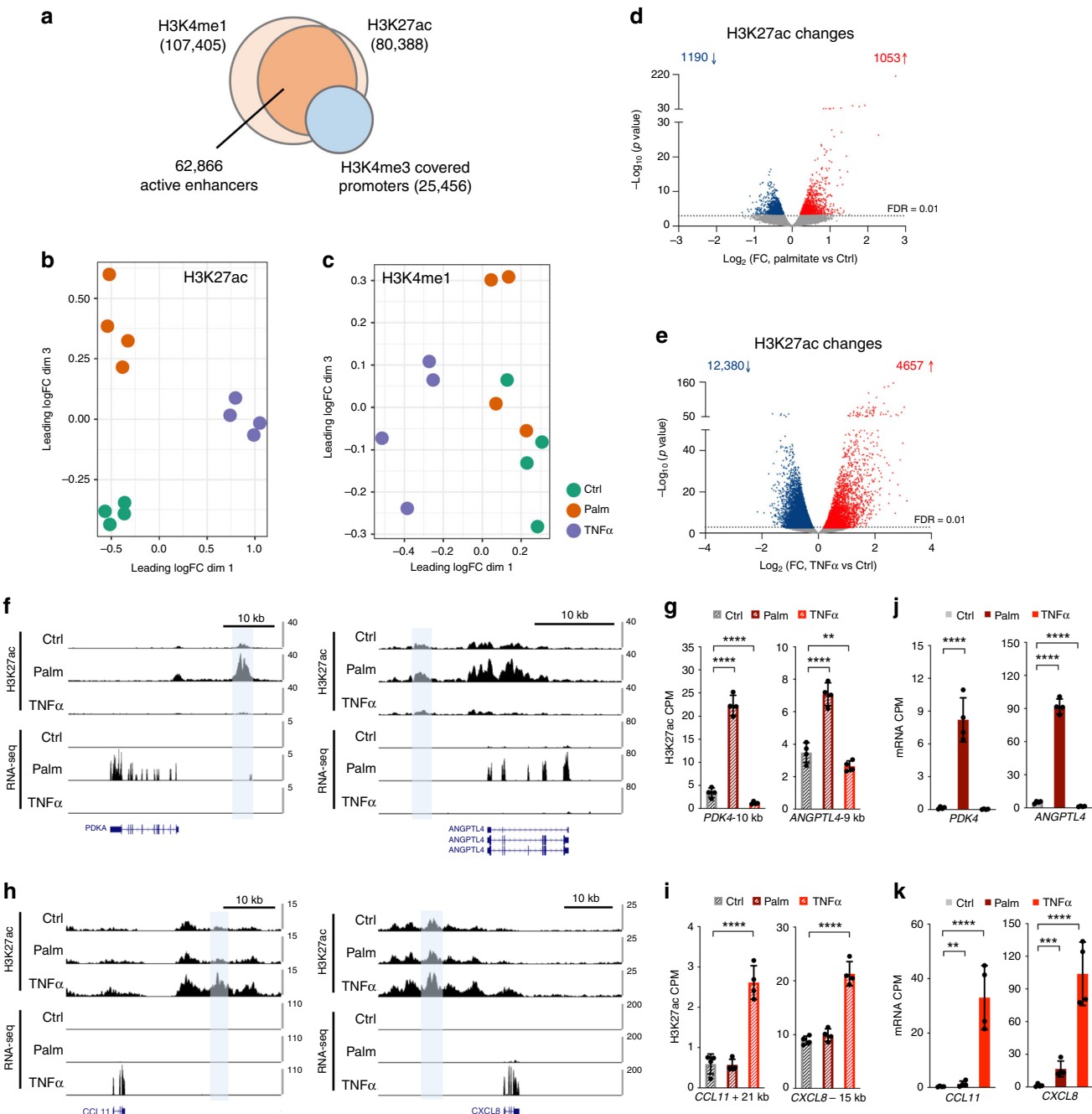

**Fig. 2 Identification of enhancers by ChIP-seq. a** Overlay of H3K4me1, H3K27ac and promoter-associated H3K4me3 ChIP-seq data from human skeletal myotubes. **b, c** MDS plot of non-promoter associated H3K27ac (**b**) and H3K4me1 (**c**) ChIP-seq data from control (ctrl), palmitate (palm) or TNFα treated cells. Leading log fold-change (logFC) is the mean logFC between the 500 most divergent H3K27ac (**b**) or H3K4me1 (**c**) ChIP-seq peaks between each pair of samples. **d, e** Volcano plot representation of differentially H3K27 acetylated regions among the 62,866 enhancers containing both H3K4me1 and H3K27ac (n = 4 biological replicates, FDR < 0.01) from palmitate (**d**) or TNFα (**e**) treated cells. Blue dots represent enhancers that are downregulated and red dots represent enhancers that are upregulated by the respective treatments (n = 4 biological replicates, FDR < 0.01). The ChIP-seq and enhancer analyses are described in detail in the Methods section. **f** and **h**, UCSC genome browser (hg38) H3K27ac and RNA-seq tracks from control (ctrl), palmitate (palm) or TNFα treated cells around *PDK4* and *ANGPTL4* (**g**) or *CCL11* and *CXCL8* (**i**). **g** and **i** Quantification of H3K27ac counts pr. million (CPM) at the selected enhancer regions in the individual replicate samples. Values are represented as the mean ± S.D. (n = 4 biological replicates). Asterisks indicate enhancers that are significantly regulated in the ChIP-seq analysis (****FDR < 0.0001, **FDR < 0.01). **j, k** Quantification of mRNA counts pr. million (CPM) of the indicated genes in the individual replicate samples. Values are represented as the mean ± S.D. (n = 4 biological replicates). Asterisks indicate genes that are significantly regulated in the RNA-seq analysis (**FDR < 0.01, ***FDR < 0.001, ****FDR < 0.0001).

confirmed the presence of H3K4me1 at these sites (Supplementary Fig. 4B, C). None of the validated enhancer regions showed enrichment of the promoter-associated H3K4me3 mark (Supplementary Fig. 4D), ruling out that these genomic regions act as

alternative promoters. Consistent with increased enhancer activity, expression of *PDK4*, *ANGPTL4*, *CCL11*, and *CCXL8* were markedly upregulated after palmitate or TNFα treatment (Fig. 2j, k), supporting a regulatory role of these enhancers on expression

of their nearby promoters. To further validate the *cis*-regulatory activity of the identified regions, we cloned the *PDK4*-10 kb and the *CCXL8*-17kb enhancers into a luciferase reporter vector. When transfected into muscle cells, luciferase activity was markedly increased in response to palmitate or TNFα treatment (Supplementary Fig. 5), confirming a regulation of enhancer activity by these treatments. Collectively, our results identify thousands of dynamic enhancer activities in human skeletal muscle cells after treatment with palmitate or TNFα.

**Capture Hi-C identifies enhancer-promoter interactions.** Besides interacting with nearby promoters, enhancers can also engage in long-range interactions, which makes enhancer-target prediction challenging. To overcome this, we performed genome-wide mapping of enhancer-promoter interactions in skeletal muscle cells by the use of high-resolution Promoter Capture Hi-C[22,24]. First, we tested if treatment of myotubes with palmitate or TNFα was associated with a dynamic reorganization of promoter-enhancer interactions. Hi-C libraries were generated from skeletal muscle myotubes followed by hybridization-based capture of 21,841 human promoters, using a collection of 37,608

biotinylated RNA baits (approximately two baits per promoter) previously designed and tested by others[22]. By sequencing the captured ligation fragments and testing for a difference in mapped Hi-C interactions by palmitate or TNFα treatment, we did not detect any significant changes (Supplementary Fig. 6A-B), suggesting that acute exposure to these treatments does not cause major changes to chromatin structure. This agrees with another study showing that TNFα-responsive enhancers are already in contact with their target promoters before transient activation or repression of enhancer activity by TNFα treatment in human fibroblasts[32].

Next, we pooled all Promoter Capture Hi-C conditions in order to obtain a general chromatin conformation capture of myotubes. This identified 36,809 significant promoter-enhancer interactions (Fig. 3a and Supplementary Data 4). Interactions covered 47% of tested promoters and 51% of identified enhancers regions (Fig. 3a) and largely spanned the entire genome (Supplementary Fig. 7). Genomic distances of identified promoter-enhancer interactions ranged up to 6.2 Mb, with a median distance of 93.8 kb (Fig. 3b) and each of the captured promoters were on average connected to 4 enhancer regions (Fig. 3c).

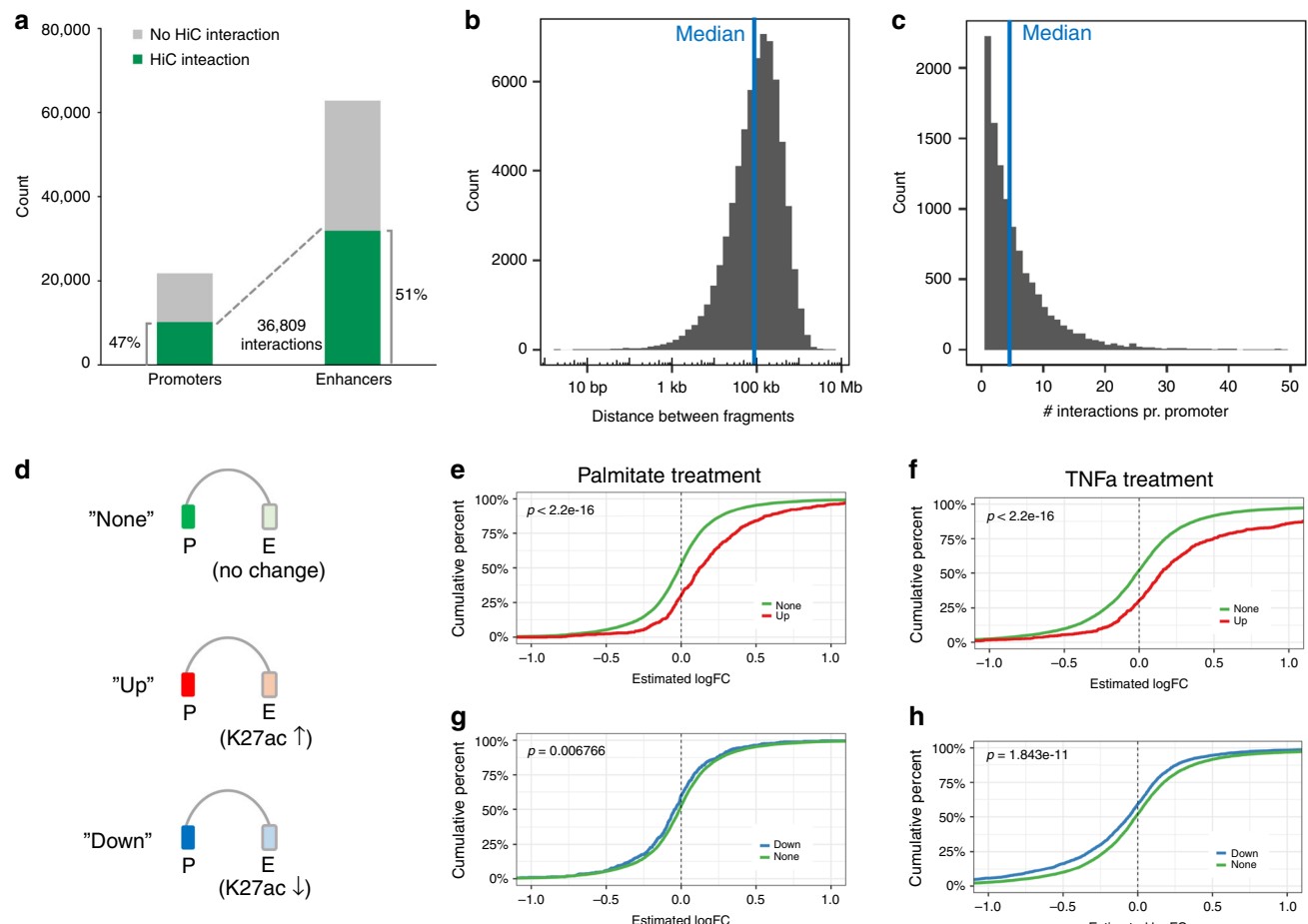

**Fig. 3 Promoter capture Hi-C identifies interactions between promoters and enhancers. a** Overview of all the significant interactions between baited promoters and H3K4me1/H3K27ac positive enhancers. **b** Histogram showing the distance between interacting promoter-fragments and enhancer-fragments. The median distance is 93.8 kb. **c** Histogram showing the number of enhancer interactions pr. promoter. The median number of interactions is 4. **d** Promoters captured by the Promoter Capture Hi-C were divided into three groups; promoters connected to enhancers that did not change H3K27ac in response to palmitate or TNFα treatment ("None"), and promoters connected to enhancers that either gained H3K27ac ("Up") or lost H3K27ac ("Down"). **e–j** Empirical cumulative distribution function (EDCF) plots of gene expression changes (RNA-seq logFC values) in the "Up" versus the "None" group for palmitate (**e**) or TNFα treatment (**f**), and the "Down" versus the "None" group for palmitate (**g**) or TNFα treatment (**h**). *X*-axis is the RNA-seq logFC, *y*-axis is the fraction of genes with this logFC or less. Differences between empirical cumulative distribution functions were tested using a Kolmogorov-Smirnov test (KS-test).

To validate if our Promoter Capture Hi-C data identified functional enhancer-promoter interactions, i.e., where a dynamic change in enhancer activity also associate with a concurrent change in promoter transcription, we divided the promoters captured in our chromatin interaction data into three groups (Fig. 3d): promoters connected to enhancers that did not change H3K27ac in response to palmitate or TNFα treatment ("None") and promoters connected to enhancers that either gained H3K27ac ("Up") or lost H3K27ac ("Down"). Empirical cumulative distribution function (EDCF) plots of gene expression changes (RNA-seq logFC values) in the different groups revealed that promoters connected to enhancers with gained activity have higher logFC values than the "None" group (Fig. 3e, f), whereas promoters connected to enhancers with decreased activity have significantly lower logFC values for both palmitate and TNFα treatments (Fig. 3g, h), supporting a regulatory role of the connected enhancers. Taken together, we have generated an enhancer-promoter connectivity map of skeletal muscle myotubes and demonstrated a general capture of promoter-enhancer pairs with concurrent changes in activity by palmitate or TNFα treatment.

**Chromatin interaction data predict enhancer target genes**. Given that the vast majority of disease-associated variants are predicted to be located in regulatory regions[9–11], our data represent an opportunity to identify target genes of GWAS SNPs in skeletal muscle cells by combining our enhancer mapping with information on chromatin conformation and gene transcription. For this, we used four sets of GWAS SNPs associated with T2D[6], IR[33–38], BMI[8] or WHR[7], as well as tagged SNPs in high linkage disequilibrium (LD, $r^2 > 0.8$) (Fig. 4a). After overlapping the variants with enhancer regions regulated by either palmitate or TNFα treatment, we identified 58 palmitate-regulated enhancers and 522 TNFα-regulated enhancers each harboring one or more GWAS SNPs (Fig. 4b). Next, we selected enhancers that were both captured by our Promoter Capture Hi-C analysis and linked to genes differentially expressed after palmitate or TNFα treatment. When only considering enhancer-gene pairs where enhancer activity and gene expression were regulated in the same direction (i.e., either upregulated or downregulated), our analysis retrieved 11 palmitate-regulated, and 124 TNFα-regulated enhancers interacting with 11 and 99 predicted target gene promoters, respectively (Fig. 4b and Supplementary Data 5). The predicted target genes included several known players in metabolism such as *IRS1*, *IGFBP3*, *PPARG*, *SOCS2*, and *LEPR*, providing a link between disease-associated SNPs and the ability of skeletal muscle to adapt to metabolic and inflammatory stress. To further narrow down the list of potential gene targets, we investigated the association between genotype of the enhancer-overlapping GWAS SNPs and the basal expression of each of their target genes in skeletal muscle biopsies of 139 individuals (by expression quantitative trait locus (eQTL) analysis). This approach identified 13 significant skeletal muscle eGenes (*CEP68*, *GAB2*, *LAMB1*, *MACF1*, *EIF6*, *PABPC4*, *BTBD1*, *FILIP1L*, *TCEA3*, *NRP1*, *ZHX3*, *TBX15*, and *TNFAIP8*) for 61 GWAS-SNPs, located within 20 distinct enhancer regions (Fig. 4c, d and Supplementary Data 6). Thus, by overlapping our genomic datasets, we have identified numerous putative target genes of metabolic GWAS SNPs, which may play a functional role under lipid toxicity or in response to proinflammatory stimuli. Moreover, for 13 genes, we demonstrate a significant association between GWAS SNP genotype and basal gene expression levels in human skeletal muscle.

**Identified target genes are linked to energy metabolism**. In order to understand the role of the identified putative GWAS-

SNP target genes in whole body metabolism in vivo, we analyzed the association between 48 metabolic traits in the BXD murine genetic reference population fed a control diet (CD) or high fat diet (HFD)[39–41] (Supplementary Data 7), and expression levels of the 13 identified eGenes in skeletal muscle (Supplementary Data 8), adipose tissue (Supplementary Data 9) and liver (Supplementary Data 10). Strikingly, expression of 12 out of the 13 genes (*Cep68*, *Gab2*, *Lamb1*, *Macf1*, *Eif6*, *Btbd1*, *Filip1l*, *Tcea3*, *Nrp1*, *Zhx3*, *Tbx15*, and *Tnfaip8*) showed associations with metabolic measures, such as blood glucose levels during glucose tolerance tests (GTTs), plasma lipid levels, body composition, and exercise performance, in at least one of the tested tissues (Table 1). For some target genes, metabolic measurements were specifically associated with expression in skeletal muscle. For example, expression of *Tbx15* (Fig. 5a), which we found linked to SNPs associated with WHR in humans, was positively associated with lean body mass (Fig. 5b) and VO$_2$ max (Fig. 5c), as well as negatively associated with total body fat mass (Fig. 5d) and blood glucose levels during an oral GTT (Fig. 5e) in the BXD mice. Interestingly, the expression of *Cep68*, which we find linked to SNPs associated with T2D, was correlated with blood glucose levels during GTTs in HFD-fed mice in both muscle and liver (Fig. 5f). More specifically, *Cep68* expression was negatively correlated with blood glucose levels during an intraperitoneal GTT in skeletal muscle of both male (Fig. 5g) and female (Fig. 5h) mice, and oral GTT in liver tissue (Fig. 5i). Moreover, *Cep68* association with body fat mass and lean mass percentages in adipose tissue (Fig. 5j) suggests that *CEP68* has a role in T2D through dysregulated expression in multiple organs. Collectively, these data demonstrate that the expression of identified putative GWAS SNP targets correlates with metabolic measures in mice, and suggest a role for these genes in the regulation of energy metabolism in vivo.

**Long-range interactions connect WHR SNPs to *EIF6* expression**. For some candidate genes identified as regulated by non-coding GWAS SNPs, including *EIF6*, the gene was not located at close vicinity of the differentially activated enhancer region, but connected through long-range chromatin interactions. The SNPs that we found linked to *EIF6* are located within the *UQCC1* locus and associate with WHR (Fig. 6a). We identified four enhancer regions, *UQCC1* + 100 kb, *UQCC1* + 26 kb, *UQCC1* + 16 kb, and *UQCC1* + 13 kb, that were all regulated by TNFα (Fig. 6b) and captured by our Promoter Capture Hi-C data. The enhancer regions overlapped several highly linked WHR-associated SNPs (Fig. 6a). From our chromatin interaction data, we found all enhancers to interact with the promoter of *EIF6* (Fig. 6a). Moreover, the *UQCC1* + 100 kb enhancer also interacted with *MMP24* and *EDEM2*, whereas *UQCC1* + 26 kb, *UQCC1* + 16 kb, and *UQCC1* + 13 kb enhancer regions looped to the *GDF5*/*CEP250* shared promoter (Fig. 6a). Out of these genes, *MMP24*, *EIF6* and *GDF5* remained candidates to be under the regulation of the enhancers, since the expression of these genes was concurrently decreased by TNFα treatment (Fig. 6c). Importantly, the *UQCC1* promoter was not found linked to the enhancer nor did *UQCC1* change expression by TNFα. While *GDF5* expression was below detection limit in skeletal muscle and could not be analyzed for eQTLs, we found associations of several LD-linked WHR associated SNPs, including rs878639, with the expression of *EIF6* (Supplementary Data 6 and Fig. 6d), but not with *MMP24* (Fig. 6e). In the case of rs878639, the major allele associates with an increased WHR, which establishes a link between lower *EIF6* expression and an unhealthy body fat distribution. Consistently, we found that *Eif6* expression in muscle from BXD mice positively associates with running distance

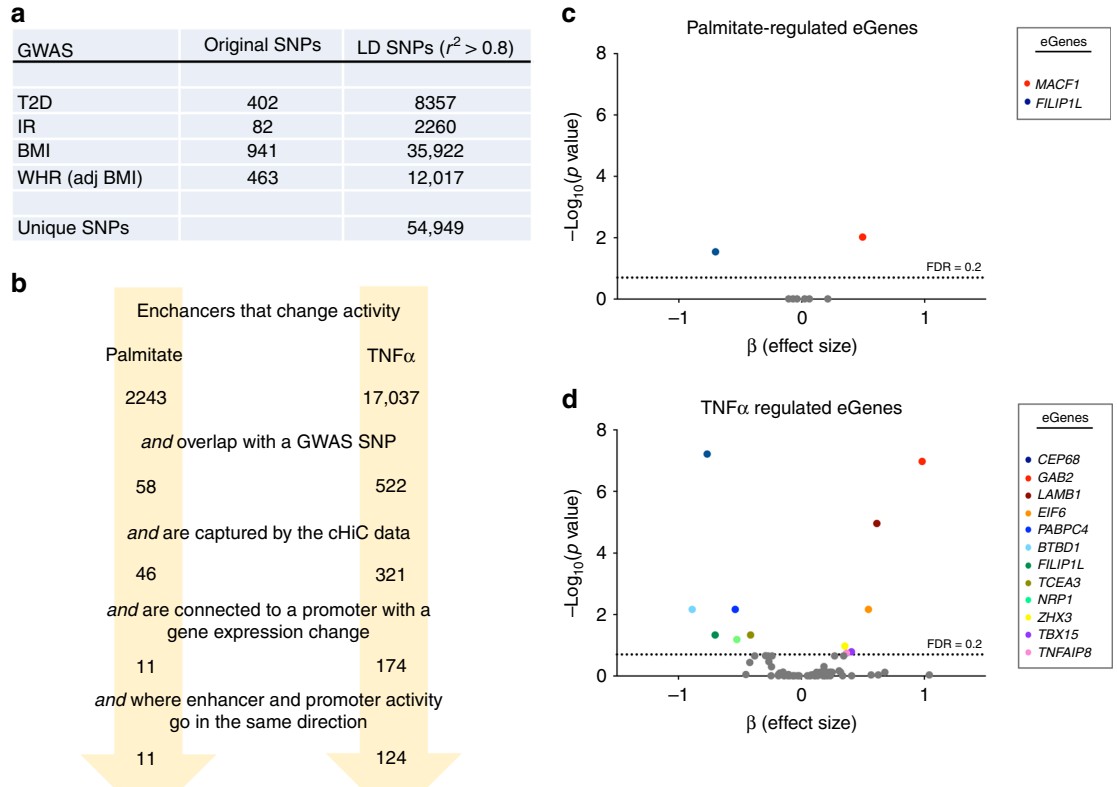

**Fig. 4 Using chromatin interaction data to predict enhancer target genes. a** Overview of the number of original and LD linked T2D, IR, BMI, or WHR GWAS SNPs. **b** Overlapping of 2243 and 17,037 palmitate-regulated or TNFα-regulated enhancers with selected GWAS SNPs, and integrating Promoter Capture Hi-C and gene expression data identifies 11 and 124 palmitate-regulated or TNFα-regulated enhancers encompassing GWAS SNPs and connected to a predicted target gene. **c, d** Volcano plot representation of eQTL analysis, where 13 significant eGenes were identified (FDR < 0.2) in total for palmitate-regulated (**c**) or TNFα-regulated genes (**d**). See also Supplementary Data 6 and the Methods section for a detailed description of the analysis.

(Fig. 6f, Supplementary Data 8), VO$_2$ basal (Fig. 6g, Supplementary Data 8) and VO$_2$ max levels after training (Fig. 6h, Supplementary Data 8), suggesting better aerobic capacity in animals with higher skeletal muscle *Eif6* expression. To further validate our findings, we used siRNAs (siEif6#1 and siEif6#2) to knock down *Eif6* expression in skeletal muscle cells (Fig. 6i and Fig. S8A). We assessed mitochondrial respiration by measuring oxygen consumption rate (OCR) at basal state or during FCCP-induced uncoupling (Fig. 6j and Supplementary Fig. 8B) and found that decreased *Eif6* expression resulted in lower OCR (Fig. 6k), especially during maximal FCCP-induced respiration (Fig. 6l and Supplementary Fig. 8C). Moreover, after differentiating C2C12 cells into myotubes, we found that *Eif6* knock-down (Supplementary Fig. 9A) led to reduced protein levels of the mitochondrial oxidative phosphorylation complex II (Fig. 6m and Supplementary Fig. 9B), whereas we did not detect any changes in insulin-stimulated glucose uptake (Supplementary Fig. 9C), glycogen synthesis (Supplementary Fig. 9D), or AKT phosphorylation (Supplementary Fig. 9E, F).

Thus, long-distance interactions networks suggest that *EIF6* is regulated by genetic variants associated with body fat distribution. Accordingly, we identified correlations between lower skeletal muscle *Eif6* expression and reduced exercise performance, and further provide evidence for a role of EIF6 in the regulation of mitochondrial function in skeletal muscle.

## Discussion
Here, we mapped the transcriptome and enhancerome of human skeletal muscle cells subjected to lipid-induced toxicity or a proinflammatory cytokine. We demonstrate a profound transcriptional reprogramming with thousands of promoter and enhancer regions showing altered activity. Integrating these data with GWAS of T2D, IR, BMI and WHR measures as well as genome-wide chromatin interaction studies, allowed us to detect concurrent changes in the activity of enhancers encompassing GWAS SNPs and transcription from a connected promoter, thereby establishing links between numerous non-coding disease-associated SNPs and gene targets. Using the murine BXD genetic reference population we provide further insight into the role of the identified target genes in the regulation of metabolic phenotypes like body composition, glucose response and exercise performance in vivo. In particular, we provide evidence that one of our identified targets, *Eif6*, controls mitochondrial respiration in skeletal muscle cells.

Our cell-system using chronic exposure with palmitate or TNFα in human primary muscle cells allowed investigation into the distinct mechanisms by which the metabolic function of the skeletal muscle cell is impaired. Palmitate induces insulin resistance at the level of AKT phosphorylation[42], impairs mitochondrial function[43], lowers expression of the master regulator of mitochondrial function peroxisome proliferator-activated receptor-gamma coactivator (PGC)-1 α[44], and induces ER stress[45]. Interestingly, incubation of skeletal muscle cells with palmitate induces TNFα secretion by the muscle cell, suggesting that while saturated fatty acids and TNFα appear to activate distinct intracellular pathways, these pathways may share common nodes[46]. Saturated free fatty acid and TNFα treatment both alter upstream insulin signaling, but TNFα treatment does not alter insulin-stimulated

**Table 1 Correlations between gene expression and metabolic phenotypes in BXD mice.**

| Gene name | Skeletal muscle | Adipose | Liver |
|---|---|---|---|
| Cep68 | Blood glucose levels | Body composition<br>Exercise performance | Blood glucose levels |
| Gab2 | None | Blood glucose levels<br>Blood insulin levels<br>Plasma lipid levels<br>Body composition<br>Exercise performance | None |
| Lamb1 | Blood glucose levels<br>Plasma lipid levels<br>Exercise performance | Plasma lipid levels<br>Body composition | None |
| Macf1 | None | N.D. | Blood glucose levels |
| Eif6 | Exercise performance | Life span | Plasma lipid levels |
| Pabpc4 | None | None | None |
| Btbd1 | Blood glucose levels<br>Exercise performance | None | Exercise performance |
| Filip1l | None | None | Blood glucose levels<br>Plasma lipid levels |
| Tcea3 | Blood glucose levels<br>Body composition<br>Exercise performance | None | Blood glucose levels<br>Body composition<br>Exercise performance |
| Nrp1 | None | Blood glucose levels | Body composition<br>Exercise performance |
| Zhx3 | None | Blood insulin levels<br>Body composition<br>Exercise performance | None |
| Tbx15 | Blood glucose levels<br>Plasma lipid levels<br>Body composition<br>Exercise performance | None | None |
| Tnfaip8 | Blood glucose levels | Body composition | None |

Overview of significant correlations for Cep68, Gab2, Lamb1, Macf1, Eif6, Pabpc4, Btbd1, Filip1l, Tcea3, Nrp1, Zhx3, Tbx15, and Tnfaip8 in skeletal muscle, adipose or liver from BXD mice (see Supplemental Tables 7–10 for more information). Macf1 expression in adipose tissue was not detected (N.D.).

glucose uptake in muscle cells whereas palmitate does[42,47]. In vivo however, TNFα infusion is associated with both lower activation of the upstream insulin-signal pathway and impaired glucose transport[48]. Even though TNFα exposure is not associated with lower fatty acid oxidation in muscle ex vivo[49], we identified EIF6 as a gene regulated by TNFα exposure and show EIF6 plays a role in fatty-acid oxidation. The discrepancy between the effects of palmitate and TNFα on primary skeletal muscle cell cultures compared to in vivo may be due to specific tissue-culture conditions, different extracellular milieus or the influence of systemic factors.

While the activity of enhancers and promoters were markedly changed after palmitate or TNFα exposure, promoter-enhancer interactions did not appear to be affected. These findings are consistent with a previous study showing that enhancers-promoter interactions are unchanged in fibroblasts treated with TNFα[32]. We cannot rule out, however, that palmitate or TNFα exposure could remodel chromatin in myotubes, as low sequencing depth or low power may have limited our capacity to detect subtle changes. From previous studies it seems clear that dynamic remodeling of promoter-enhancer interactions occurs during cellular differentiation, particularity at cell type-specific enhancers[23,50–53]. Interestingly, the discrepancy between activation of cell type-specific enhancers and enhancers induced by treatments such as TNFα seems to correlate with H3K4me1 levels. Indeed, treatment-induced enhancers appear to exhibit largely unchanged levels of H3K4me1, despite a quick induction of H3K27ac, whereas cell type-specific enhancers display highly variable H3K4me1 levels[32]. This is consistent with our data, where palmitate- and TNFα-induce large changes in H3K27ac

levels at enhancers but only minor changes in H3K4me1. Still, certain chromatin interactions were recently described to be variable in a circadian fashion[54], suggesting that promoter-enhancer interactions can indeed be dynamic even within a defined cell type.

Our mapping of the chromatin interactome of human myotubes identified 36,809 specific enhancer-promoter interactions. Integrating these data with RNA transcription and enhancer activity analyses allowed us to specifically capture enhancer-promoter interactions where 1) the enhancer overlaps one or more SNPs associated with T2D, IR, BMI or WHR and 2) the enhancer activity and gene expression were regulated in the same direction by either palmitate or TNFα exposure. Our analysis retrieved more than 100 predicted GWAS target genes, which included several known players in metabolism such as IRS1, IGFBP3, PPARG, SOCS2, and LEPR. However, our eQTL analysis did not detect an association between genotype and gene expression for most of these genes. We therefore speculate that GWAS SNPs may be functionally linked with gene expression in situations of cellular stress encountered in metabolic disease such as increased plasma levels of fatty acids or proinflammatory cytokines.

For the genes identified as significant eGenes in our eQTL analysis, we analyzed the association between their expression levels in skeletal muscle, adipose, or liver tissue and measures of 48 metabolic traits in the BXD murine genetic reference population. We found that 12 out of 13 genes (Cep68, Gab2, Lamb1, Macf1, Eif6, Btbd1, Filip1l, Tcea3, Nrp1, Zhx3, Tbx15, and Tnfaip8) exhibited marked associations with metabolic phenotypes in one or more of the tested tissues. For some targets,

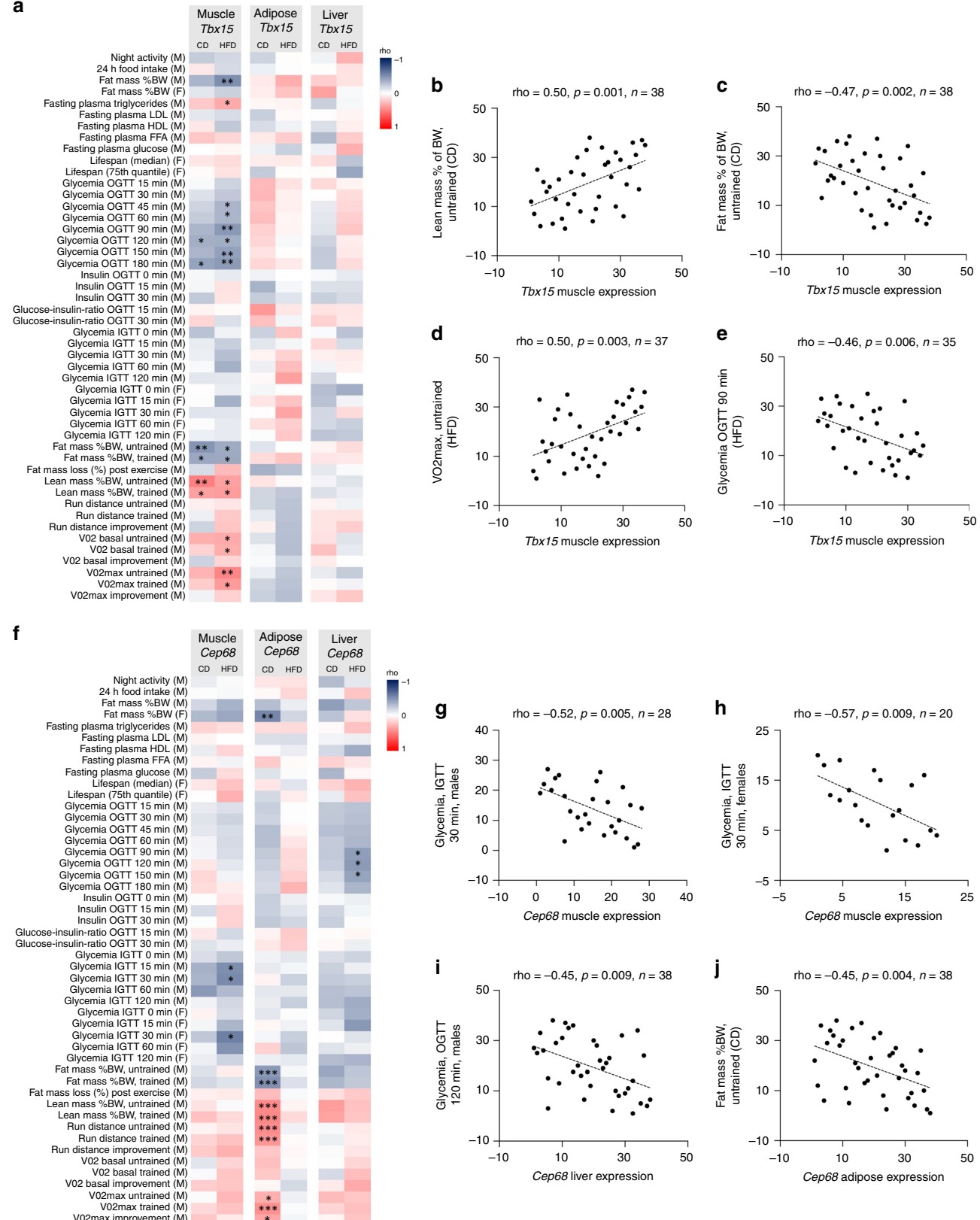

including *Tbx15*, the associations appeared specific for skeletal muscle expression and were not detected in either adipose or liver tissue, suggesting a muscle-specific role of *Tbx15*. This is consistent with the earlier finding that *Tbx15* regulates muscle metabolism in mice and *Tbx15* knockout animals are resistant to diet induced obesity and impaired glucose tolerance[55]. For other targets, such as *Cep68*, we identified associations in all of the tested tissues revealing the metabolic role of these genes in multiple organs. Linking gene expression with metabolic phenotypes represents a valuable tool to gain insight into gene function,

**Fig. 5 Correlating GWAS SNP-target genes with metabolic phenotypes in BXD mice strains. a** Heatmap representation of rho-values from correlations between 48 metabolic measurements in CD or HFD fed mice and *Tbx15* expression in skeletal muscle, adipose or liver tissue. The *p*-values from the 48 correlations from each diet and tissue were adjusted using false discovery rate correction (FDR) (*FDR < 0.2, **FDR < 0.1, ***FDR < 0.05). **b–e** Skeletal muscle expression of *Tbx15* is positively correlated with lean mass (% of body weight) (**b**), negatively correlated with fat mass (% of body weight) (**c**) positively correlated with VO2$_{max}$ (**d**) and negatively correlated with glycemia during an oral GTT (OGTT) (**e**). Statistics was performed using Spearmans rank correlation analysis. **f** Heatmap representation of rho-values from correlations between 48 metabolic measures in CD or HFD fed mice and *Cep68* expression in skeletal muscle, adipose or liver tissue (*FDR < 0.2, **FDR < 0.1, ***FDR < 0.05). **g–j** *Cep68* is negatively correlated with glycemia during an intraperitoneal GTT (IGTT) in both male (**g**) and female (**h**) mice in skeletal muscle, as well as an oral GTT (OGTT) in liver (**i**) and fat mass (% of body weight) in adipose tissue (**j**). Statistics was performed using Spearmans rank correlation analysis.

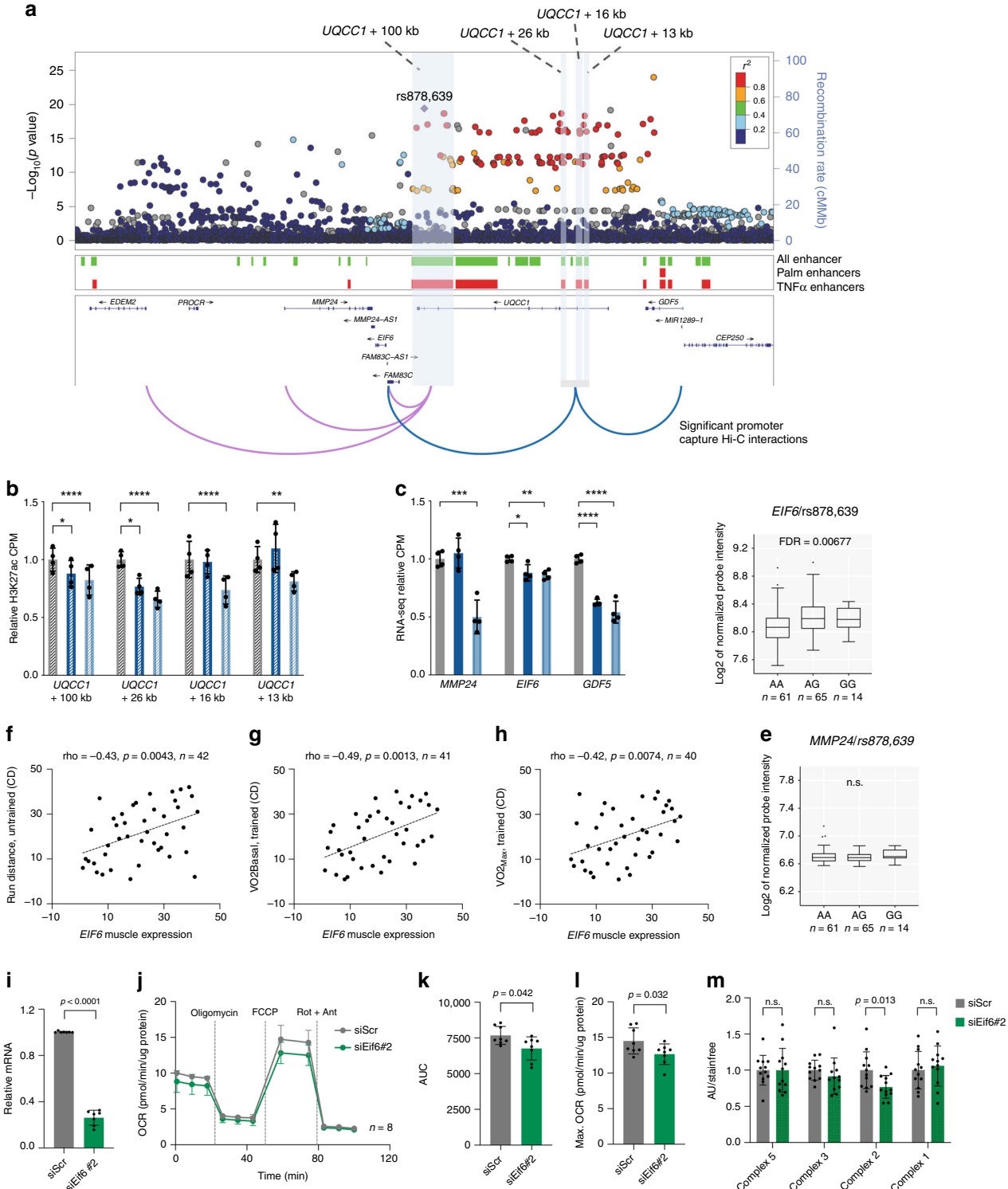

**Fig. 6 Long-range interactions connect WHR-associated rs878639 to *EIF6* expression. a** Regional visualization of WHR GWAS data[7] at the region around *UQCC1* with highlight of rs878639 and linked SNPs. Position of all enhancers (green), palmitate or TNFα regulated enhancers (red) are indicated below. Identified Promoter Capture Hi-C interactions for the *UQCC1* + 100 kb (pink) and the *UQCC1* + 26 kb, *UQCC1* + 16 kb, and *UQCC1* + 13 kb enhancer region (blue) are illustrated. **b** Quantification of H3K27ac counts pr. million at the *UQCC1* + 100 kb and *UQCC1* + 26 kb, *UQCC1* + 16 kb, and *UQCC1* + 13 kb enhancers from control, palmitate-treated or TNFα-treated cells. Values are represented as mean ± S.D. ($n = 4$ biological replicates). **c**, Quantification of *MMP24*, *EIF6* and *GDF5* RNA-seq counts pr. million from control, palmitate-treated or TNFα-treated cells. Values are represented as mean ± S.D. ($n = 4$ biological replicates). For **b**, **c**; Stars indicate enhancers or genes that are significantly regulated in the ChIP-seq or RNA-seq analyses (****FDR < 0.0001, ***FDR < 0.001, **FDR < 0.01, *FDR < 0.05). **d**, **e** eQTL analysis in skeletal muscle between rs878639 and *EIF6* (**d**) or *MMP24* (**e**) expression. Data are presented as box plots where the horizontal line represent the median, vertical middle bars represent the first and third quartiles, and black dots denote outliers beyond 1.5 times the interquartile range (Tukey plot). **f**–**h** Skeletal muscle expression of *Eif6* is positively correlated with running distance (**f**), VO2$_{basal}$ (**g**) and VO2$_{max}$ (**h**) in BXD mice strains. Statistics were performed using Spearmans rank correlation analysis. **i**, *Eif6* mRNA levels in siScr or siEif6#2 transfected C2C12 myoblasts. Expression data was normalized to housekeeping *Gapdh* expression levels. Values are represented as mean ± S.D. ($n = 7$) **J**, Real-time measurements of oxygen consumption rates (OCR) by Seahorse Extracellular Flux Analyzer in siScr or siEif6#2 transfected C2C12 cells. OCR was measured under basal conditions and after injection of oligomycin, FCCP, and antimycin A combined with rotenone at indicated time points. Values are represented as mean ± S.D. ($n = 8$). **k**–**l** OCR area under the curve (AUC) values (**k**) or mean OCR for the time points during FCCP-induced maximal respiration (**l**) for siScr or siEif6#2 transfected C2C12 myoblasts. Values are represented as mean ± S.D. ($n = 8$) **m**, Quantification of Western blots of the mitochondrial oxidative complexes V, III, II and I in siScr or siEif6#2 transfected C2C12 cells. Values are represented as mean ± S.D. ($n = 12$), statistical tests were performed by a two-tailed *t*-test (n.s., *p* > 0.05).

although it does not infer on causality. Circulating leptin levels, for instance, are positively associated with fat mass[56], but loss-of-function mutations of *LEP* are associated with obesity[57]. In our study, we observed a similar phenomenon where the *CEP68* T2D risk variants are associated with increased *CEP68* expression, but *Cep68* expression is negatively associated with blood glucose levels during GTTs in mice. While further investigations are warranted to establish causal relationships and the mechanism by which *CEP68* may regulate whole body metabolism, we speculate that dysregulated expression of *CEP68* is involved in the pathogenesis of T2D.

For some genes that we identified as potential targets of metabolic GWAS SNPs, the SNP-enhancer locus was not located in close proximity to the predicted target gene, but engaged in long-range DNA looping formations. For example, we identified interactions between the promoter of the translation initiation factor *EIF6* and several enhancers located within the *UQCC1* gene, each spanning SNPs associated with WHR in humans. We found both enhancers and *EIF6* expression were downregulated by TNFα and we detected significant eQTLs for *EIF6* expression with SNPs of all loci. In the BXD mice, *Eif6* muscle expression was associated with increased running distance, as well as with basal and maximal VO2 uptake after training. These findings are consistent with a study linking *EIF6* to the regulation of energy metabolism during endurance training in humans and showing reduced exercise performance in *Eif6* haploinsufficient mice[58]. Moreover, hypermethylation of the *EIF6* promoter is linked to childhood obesity[59]. In support of this, we demonstrate that *Eif6* knockdown in murine muscle cells causes lower mitochondrial respiration and reduced levels of the mitochondrial oxidative complex II. The identified link between *EIF6* and modulation of WHR are consistent with data demonstrating that genetic variants within mitochondrial genes are associated with metabolic measures including WHR[60]. Notably, we did not detect a physical link between the *UQCC1* intronic enhancers and the *UQCC1*-promoter, nor did *UQCC1* change expression by TNFα. A recent study has shown that human *UQCC1* coding variants are associated with WHR[61]. Interestingly, eQTL analysis indicates that these variants associate not only with the expression levels of *UQCC1*, but also *EIF6*[61], suggesting that several genes within this locus could contribute to the modulation of WHR in humans. Thus, our data demonstrate that *EIF6* expression is regulated by TNFα and suggest a role for muscle-specific expression of *Eif6/EIF6* in the regulation of mitochondrial function and exercise performance in mice, as well as in WHR ratio in humans.

In conclusion, our study identified skeletal muscle enhancer elements that are dysregulated in the context of lipid-toxicity or under exposure of the proinflammatory cytokine TNFα. We identify hundreds of dysregulated enhancers which overlap with genetic loci previously implicated in metabolic disease and, using chromatin conformation assay, we predict the corresponding gene targets. We identify genes with known roles in metabolism, as well as targets that have not previously been linked to human metabolic disease, and demonstrate their association with metabolic phenotypes in mice. Given the influence of lifestyle and genetic factors in the development of obesity and T2D, and the prominent contribution of skeletal muscle in energy metabolism in humans, our investigations constitute a resource for identifying genes participating in the progression of metabolic disorders.

## Methods

**Cell culture**. Human skeletal muscle cells (CC-2561 from Lonza) were cultured in DMEM/F-12, GlutaMAX™ (Life Technologies) supplemented with 20% FBS (Sigma-Aldrich) and 1% penicillin/streptomycin (Life Technologies) during proliferation. Differentiation was initiated when cells were 80% confluent by addition of differentiation media (DMEM/F-12, GlutaMAX™ supplemented with 2% FBS (Sigma-Aldrich) and 1% penicillin/streptomycin). Cells were differentiated for 5–7 days. For palmitate and TNFα treatment, the differentiated myotubes were added 0.5 µM palmitate for 48 h (on day 5–7 of differentiation) or 10 ng/ml TNFα for 24 h (on day 6–7 of differentiation).

Murine C2C12 myoblasts (CRL-1772 from ATCC) were cultured in DMEM (Life Technologies) supplemented with 10% FBS (Sigma-Aldrich) and 1% penicillin/streptomycin (Life Technologies). Differentiation was initiated when cells were 80% confluent by addition of differentiation media (DMEM/F-12, GlutaMAX™ supplemented with 2% horse serum (Sigma-Aldrich) and 1% penicillin/streptomycin). Cells were differentiated for 5 days.

Insulin stimulation experiments for human skeletal muscle cells and C2C12 cells were performed by serum depriving differentiated myotubes for 4 h before stimulating with either 10 or 100 nmol/L insulin for 5 min.

**Western blotting**. For Western blotting, protein lysates were prepared using a phospho-protein lysis buffer (20 mM Tris (pH 7.4), 150 mM NaCl, 5 mM EDTA, 150 mM NaF, 2 mM Na$_3$VO$_4$, 10 mM sodium pyrophosphate, 0.5 mM phenylmethylsulfonyl fluoride). Immunoblotting was performed according to standard protocols using total-AKT (CST-9272S, 1:1000 dilution), Phospho-Ser473 AKT (CST-9271S, 1:1000 dilution), or OXPHOS cocktail (Abcam 110413, 1:5000 dilution) as primary antibodies and goat anti-rabbit (Bio-Rad 170-6515, 1:10,000 dilution) or goat anti-mouse IgG (Bio-Rad 170-6516, 1:10,000 dilution) horseradish peroxidase conjugate secondary antibodies. Total protein on the membrane was quantified using Bio-Rad stain-free gels. Images were developed with ImmunStar WesternC Chemiluminescence kit (Bio-Rad) using a Molecular Imager ChemiDoc XRS + (Bio-Rad) and analyzed using ImageLab software (Bio-Rad).

**Luciferase assays**. Genomic DNA was purified from human skeletal muscle cells using DNeasy Blood & Tissue Kit (Qiagen). The *PDK4*-10 kb (hg38, chr7:95,608,432-95,609,655) and *CXCL8*-17kb (hg38, chr4:73,725,619-73,726,851)

enhancers were PCR amplified (primers are listed in in Supplementary Data 12) from skeletal muscle DNA and ligated into the pGL4.23[luc2/minP] vector (Promega) using KpnI and NheI restriction enzymes for the *PDK4*-10kb and NheI and XhoI for *CXCL8*-17kb. All constructs were verified by Sanger sequencing. pGL4.23 [luc2/minP] plasmids, either empty or containing enhancer fragments, were co-transfected with 1:100 pRL-SV40 vector (Promega) into skeletal muscle cells in 96-well plates using TransIT-X2 (Mirius) according to manufactures protocol ($n = 5$ independent transfections). After 24 h, cell media was changed and cells to be treated with palmitate or TNFα were added 0.5 mM palmitate or 10 ng/ml TNFα, respectively. After 48 h, luminescence was determined using the Dual-Glo Luciferase Assay System (Promega) and a Hidex Sense microplate reader. Firefly luciferase counts were normalized to Renilla luciferase counts.

**siRNA mediated knockdown of *Eif6*.** C2C12 myoblasts were reverse transfected by seeding cells together with transfection mix containing siScr (SIC001-10NMOL, Sigma-Aldrich) or siRNAs against *Eif6* using siEif6#1 (SASI_Mm01_00034707: 5′-cucuggacuuuggcucauu-3′, Sigma-Aldrich) or siEif6#2 (SASI_Mm01_00034708: 5′-gucagagcgucguucgaga-3′, Sigma-Aldrich). Transfections were performed using TransIT-X2 (Mirius) according to manufactures protocol and cells were assayed 48 h after transfection (myoblasts) or 5 days after differentiation (myotubes).

**Measurement of oxygen consumption rate (OCR).** Real-time measurements of OCR were performed using a Seahorse XFe96 Extracellular Flux Analyzer (Agilent Technologies). C2C12 myoblasts were reverse transfected by seeding 5000 cells per well in Seahorse XFe96 Cell Culture Microplates (Agilent Technologies) together with transfection mix containing siScr or siRNAs against *Eif6* (siEif6#1, $n = 6$ biological replicates or siEif6#2, $n = 8$ biological replicates). Cells were assayed 48 h after transfection using the Seahorse XF Cell Mito Stress Test kit (Agilent Technologies). OCR was measured under basal conditions and after injection of final concentrations of 1 μM oligomycin, 2.3 μM FCCP, or 2.55 μM antimycin A combined with 1 μM rotenone. The measured OCR values were normalized to protein levels by lysing the cells and performing BCA protein assay (Pierce BCA Protein Assay Kit from Thermo Scientific).

**Glucose uptake.** Differentiated myotubes in 12-well plates were serum-starved for 4 h, washed with PBS then stimulated in the presence or absence of 10 nM insulin in 500 μl of KRP buffer (pH 7.3) for 20 min. 25 μl of 0.2 mM 2-deoxy-glucose, 10 μCi/ml [1,2-3 H (N)] 2-deoxy-glucose (Perkin Elmer) was added for the last 5 min of insulin stimulation and then cells were washed 3 times with cold PBS on ice and cells were lysed in 200 μl of phospho-protein lysis buffer. Radioactivity was determined by liquid scintillation counting after the addition of Ultima Gold LSC. Values were subtracted from background samples for each condition (cells treated with 2 μl of 10 mM Cytochalasin B during the insulin stimulation). Values were normalized to protein levels performing BCA protein assay (Pierce BCA Protein Assay Kit from Thermo Scientific).

**Glycogen synthesis.** Differentiated myotubes in 12-well plates were serum-starved for 4 h, followed by a 1 h incubation in KRP buffer (pH 7.3) containing 5 mM glucose, 2 μCi/ml Glucose, D-[U-14C] (Perkin Elmer), in the absence or presence of 100 nM insulin. Cells were washed 3 times in cold PBS, harvested in 200 μl of 1 M NaOH and heated to 70 °C for 15 min. Ten microliter was taken for the determination of protein (BCA) and to the remainder, 25 μl saturated $Na_2SO_4$, and 900 μl ice-cold ethanol was added, vortexed and frozen for 30 min at −80 °C, followed by a centrifugation step (10 min, 16,000×g, 4 °C). Pellets were resuspended in 100 μl $H_2O$, followed by addition of 1 ml ice-cold ethanol and re-centrifugation. The final pellet was resuspended in 100 μl $H_2O$ and radioactivity was determined by liquid scintillation counting after the addition of Ultima Gold LSC. Values were normalized to protein levels performing BCA protein assay (Pierce BCA Protein Assay Kit from Thermo Scientific).

**RNA purification.** Total RNA was purified from human skeletal myotubes (control, palmitate or TNFα treated) using AllPrep DNA/RNA/miRNA Universal Kit (Qiagen). For quantification, total RNA was reverse-transcribed using iScript™ cDNA Synthesis Kit (Bio-Rad), according to the manufacturer's instructions and analyzed by real-time PCR using Brilliant III Ultra-fast SYBR Green QPCR Master Mix (AH Diagnostic) and a C1000 Thermal cycler (Bio-Rad). mRNA primer sequences are listed in Supplementary Data 12.

**RNA-sequencing.** One microgram of total RNA was depleted of rRNA and subsequently used to generate libraries using the TruSeq standard total RNA with Ribo-Zero Gold kit (Illumina). The PCR cycle number for each library amplification was optimized by running 10% of the library DNA in a real-time PCR reaction using Brilliant III Ultra-fast SYBR Green QPCR Master Mix (AH Diagnostic) on a C1000 Thermal cycler (Bio-Rad) (Supplementary Data 11). Libraries were sequencing on a NextSeq500 system (Illumina) using the NextSeq 500/550 High Output v2 kit (75 cycles). An overview of all RNA-seq experiments are given in Supplementary Data 11.

For bioinformatic analysis of RNA-seq data, reads were aligned to the hg38 GENCODE Comprehensive gene annotations[62] version 27 using STAR v2.5.3a[63]. Read summation onto genes was performed by featureCounts v1.5.3[64]. Differential expression testing was performed with edgeR v3.14.0[65] using a model of the form ~0 + group + block, where group was a factor containing information on both passage and treatment, and block encoded the two replicates. Differential expression was found by testing e.g., (P5_Palmitate + P6_Palmitate)/2 – (P5_Control + P6_Control)/2 using the quasi-likelihood tests in edgeR. GO enrichments were found using the camera function[66], which takes both inter-gene correlations and the distribution of log fold changes in the data-set into consideration and is part of the edgeR package. Only gene ontologies containing between 10 and 500 genes were investigated. Initial visualization of samples was performed by multi-dimensional scaling (MDS) plots, which are similar to PCA plots but use average log fold changes of the 500 most divergent interactions.

**ChIP-sequencing.** Skeletal muscle myotubes were treated with palmitate or TNFα ($n = 4$ biological replicates using cells from two different passages), and cross-linked in 1% formaldehyde in PBS for 10 min at room temperature followed by quenching with glycine (final concentration of 0.125 M) to stop the cross-linking reaction. Cells were washed with PBS and harvested in 1 ml SDS Buffer (50 mM Tris-HCl (pH 8), 100 mM NaCl, 5 mM EDTA (pH 8.0), 0.2% NaN₃, 0.5% SDS, 0.5 mM phenylmethylsulfonyl fluoride) and centrifuged for 6 min at 250 × g. The pelleted nuclei were lysed in 1.5 ml ice-cold IP Buffer (67 mM Tris-HCl (pH 8), 100 mM NaCl, 5 mM EDTA (pH 8.0), 0.2% NaN₃, 0.33% SDS, 1,67% Triton X-100, 0.5 mM phenylmethylsulfonylfluoride) and sonicated (Diagenode, Bioruptor) to an average length of 200–500 bp (between 15 and 20 cycles, high intensity). Before starting the ChIP experiment, chromatin was cleared by centrifugation for 30 min at 20,000 × g. For each ChIP, 2–10 μg DNA was combined with 2.5 μg antibody and incubated with rotation at 4 °C for 16 h. The following antibodies were used for ChIP: H3K27ac (Ab4729), H3K4me1 (Ab8895), H3K4me3 (CST-9751S), H3 (Ab1791). Immunoprecipitation was performed by incubation with Protein G Sepharose beads (GE healthcare) for 4 h followed by three washes with low-salt buffer (20 mM Tris-HCl (pH 8.0), 2 mM EDTA (pH 8.0), 1% Triton X-100, 0.1% SDS, 150 mM NaCl) and two washes with high-salt buffer (20 mM Tris-HCl (pH 8.0), 2 mM EDTA (pH 8.0), 1% Triton X-100, 0.1% SDS, 500 mM NaCl). Chromatin was de-cross-linked in 120 μl 1%SDS and 0.1 M NaHCO₃ for 6 h at 65 °C, and DNA was subsequently purified using Qiagen MinElute PCR purification kit. For library preparation and sequencing, 3–10 ng of immunoprecipitated DNA was used to generate adapter-ligated DNA libraries using the NEBNext® Ultra DNA library kit for Illumina (New England Biolabs, E7370L) and indexed multiplex primers for Illumina sequencing (New England Biolabs, E7335). The PCR cycle number for each library amplification was optimized by running 10% of the library DNA in a real-time PCR reaction using Brilliant III Ultra-fast SYBR Green QPCR Master Mix (AH Diagnostic) and a C1000 Thermal cycler (Bio-Rad) (Supplementary Data 11). DNA libraries were sequenced on a HiSeq2000 by 50-bp single-end sequencing at the National High-Throughput Sequencing Centre (University of Copenhagen, Denmark). An overview of all ChIP-seq experiments are given in Supplementary Data 11.

ChIP-qPCR validations were performed by ChIP followed by real-time PCR using Brilliant III Ultra-fast SYBR Green QPCR Master Mix (AH Diagnostic) and a C1000 Thermal cycler (Bio-Rad). All reactions were analyzed in quadruplicates. ChIP-qPCR primer sequences are listed in Supplementary Data 12.

For bioinformatic analysis of ChIP-seq data, sequenced reads were aligned using the sub-read aligner v1.5.0[67] against a full index of the main chromosomes of the hg38 reference genome, as genomic DNA and keeping only uniquely mapped reads. Duplicate reads were removed using Picard tools (http://broadinstitute.github.io/picard). Peaks were called using MACS2 v2.1.0.20150731[68] with input control. H3K4me1 peaks were called as broad peaks, while H3K27ac peaks were called as narrow peaks. The quality of individual samples was assessed by testing whether fragment lengths could be estimated and whether more than 200,000 peaks could be called with a *P*-value cutoff of 0.05. These individual peak lists were only used to identify samples where the IP-step had failed and were not used in the downstream analysis. All samples passed these two tests. The consensus peak list used in the analysis was generated following the ENCODE 2012 IDR pipeline. For each histone modification a consensus peak set was generated as follows. All samples were pooled and the pooled reads were shuffled and split in two (pseudo replicates). Initial peak lists were called as above on each of these three samples (pool and two pseudo replicates), with a *P*-value cutoff of 0.05 and sorted by *P*-value. Finally, a consensus peak list was generated using the irreproducible discovery rate (IDR) software v2.0.2[69] with the pseudo replicate peak lists as input and the pooled peak list as oracle peak list. The IDR is analogous to an FDR, and has been shown to be a better measure of reproducibility in peak-calling experiments[70]. A lenient IDR threshold of 0.05 was used. For each sample, reads were summarized into consensus peaks using featureCounts v1.20.6[64]. Differentially bound peaks were detected in edgeR v3.22.0 as described[65], using reads along the entire peak and the same model and testing procedure as in the RNA-seq analysis. Peaks were considered overlapping if they overlapped by any amount.

**Enhancer mapping.** H3K4me3 peaks from human skeletal muscle myotubes derived from skeletal muscle myoblasts were downloaded from Roadmap Epigenomics[71] (sample E121), lifted to hg38 using the UCSC liftOver tool[72] and filtered

to keep peaks with a FDR < 0.05. Active promoters were defined as RefSeq gene[73] promoters with a H3K4me3 peak within 3000 bp upstream or 1000 bp downstream of its TSS. Enhancers were defined as regions that contained a consensus peak of both H3K27ac and H3K4me1, as defined in the ChIP-seq, and was more than 3000 bp upstream or 1000 bp downstream of the TSS of an active promoter.

**Promoter capture Hi-C.** Two 15 cm plates of in vitro differentiated myotubes (n = 3 biological replicates using three different passages of cells) were treated with either palmitate or TNFα or left untreated as control. Promoter Capture Hi-C was performed using similar protocols as described in[22,24]. Cells were cross-linked in 2% formaldehyde for 10 min followed by quenching with glycine (final concentration of 0.125 M). After washing with PBS, the cells were centrifuged for 10 min at 400 × g and frozen at −80 °C until further analysis. Cells were lysed in 50 ml ice-cold lysis buffer (10 mM Tris-HCl pH 8, 10 mM NaCl, 0.2% Igepal CA-630 and protease inhibitor cocktail (Roche complete, EDTA-free)). After 30 min incubation on ice, nuclei were pelleted by centrifugation at 650 × g for 5 min. The pellet was resuspended in 1.25× NEBuffer 2 and added SDS (final concentration of 0.3%) followed by rotation at 37 °C for 1 h. Triton X-100 was added (final concentration of 1.7%) and the samples were incubated shaking at 37 °C for 1 h. After digesting with HindIII (NEB R0104T, 1500 units per 5 million cells starting material) at 37 °C overnight, restriction fragment overhangs were filled by Klenow (NEB) adding biotin-14-dATP (Life Technologies), dCTP, dGTP and dTTP (all at a final concentration of 30 μM) and incubating for 60 min at 37 °C. Enzymes were deactivated by adding SDS (final concentration of 1.47%) and incubated shaking for 30 min at 65 °C. Ligation was performed using 50 units T4 DNA ligase (Invitrogen) per 5 million cells starting material in a total volume of 8.2 ml 1X ligation buffer (NEB B0202S) containing 100 μg/ml BSA (NEB) and 0.9% Triton X-100, and by incubating for four hours at 16 °C followed by 30 min at room temperature. Cross-links were reversed by incubation with Proteinase K at 65 °C overnight. After 16 h, additional Proteinase K was added and the samples were further incubated for 2 h at 65 °C. RNase A treatment was performed for 60 min at 37 °C, and DNA was purified by two sequential phenol-chloroform extractions. DNA concentration was measured using a Qubit Fluorometer and Qubit dsDNA HS Assay Kit (Life technologies). In order to remove biotin from non-ligated DNA ends, 40 μg DNA was incubated with T4 DNA polymerase in a buffer containing 1× NEBuffer 2, 0.1 mg/ml BSA, and 0.1 mM dATP for 4 h at 20 °C followed by phenol-chloroform extraction. DNA was sheared by sonication (Diagenode, Biorupter) to an average length of 400 bp (20 cycles, low intensity), followed by DNA end-repair by incubation with T4 DNA polymerase (NEB M0203L), T4 Polynucleotide kinase (NEB M0201L), Klenow (NEB M0210L), and dNTP mix (0.25 mM) in 1X ligation buffer (NEB B0202S). After 30 min incubation at room temperature, DNA was purified using Qiagen MinElute PCR purification kit. For addition of dATP to the Hi-C libraries, DNA was incubated with Klenow exo- and 0.23 mM dATP in 1X NEBuffer 2 for 30 min at 37 °C. Enzymes were inactivated by incubation at 65 °C for 20 min. DNA fragments were size-selected by a double-sided SPRI bead purification (SPRI beads solution volume to sample volume to 0.6:1 followed by 0.9:1). Biotin-marked ligation products were isolated using MyOne Streptavidin C1 Dynabeads (LifeTechnologies). After washing the beads in tween buffer (5 mM Tris, 0.5 mM EDTA, 1 M NaCl, 0.05% Tween), binding of DNA was performed in binding buffer (5 mM Tris, 0.5 mM EDTA, 1 M NaCl) for 30 min at room temperature, followed by two washes in binding buffer, and one wash in ligation buffer (NEB B0202S). The beads were resuspended in ligation buffer and adapters (from SureSelect XT library prep kit ILM, Agilent Technologies) were ligated to the bead-bound DNA by the addition of T4 DNA ligase (NEB) and incubation for 2 h at room temperature. The beads were subsequently washed twice in tween buffer, once in binding buffer and twice in 1X NEBuffer 2 before resuspending the beads in 40 μl 1X NEBuffer 2. The bead-bound library DNA was amplified with 12–14 PCR amplification cycles according to the SureSelect XT library prep kit ILM (Agilent Technologies) protocol before promoter capture. Promoter capture was performed by using 37,608 biotin-labeled RNA baits (each 120 nucleotides) covering 21,841 human promoters (approximately two baits per promoter, targeting each end of a HindIII fragment[22]). The RNA baits were synthesized by Agilent Technologies and hybridization was performed using the Sure Select Target Enrichment kit ILM (Agilent Technologies) and SureSelect XT library prep kit ILM (Agilent Technologies) according to manufacturer's instructions. DNA libraries were paired-end sequenced on a NextSeq500 system (Illumina) using the NextSeq 500/550 High-Output v2.5 Kit (150 cycles).

For bioinformatic analysis of Promoter Capture Hi-C data, di-tags reads were filtered and mapped against the main chromosomes of the hg38 reference genome by the HiCUP pipeline v0.6.1[74] using bowtie2 v2.2.6[75] without limits on maximum and minimum di-tag length. The HiCUP pipeline also removes PCR duplicate reads and filters out re-ligations and other experimental artefacts. Downstream analysis was performed with diffHic[76] as follows: di-tags were filtered keeping only DNA fragments shorter than 600 bp, and with a minimum inwards and outwards facing gap distance of 1000 and 25,000, respectively, as recommended in the manual. All conditions and passages of cells were then pooled to obtain a general chromatin conformation capture of myotubes; all enhancers were widened to 10 kb and interactions between a promoter and a histone mark were extracted and filtered to remove weak interactions so that only interactions with an average mean of 5 counts per million, calculated by the aveLogCPM function, and with a signal at

least two-fold above the expected were kept (as calculated by the filterTrended function—see the diffHic manual for code examples). The connectCounts function was used to count reads supporting interactions for each library, interactions with enough reads to test for differential binding were selected using the filterByExpr function of edgeR and differential binding was performed using the edgeR quasi-likelihood function as in the RNA-seq and ChIP-seq experiments but without the replicate blocking factor, resulting in a model of the form ~0+ group. These criteria and cut-offs were as described in the diffHic package manual. The set of interactions interrogated for differential interactions is the one used in downstream analysis and reported in the Supplementary tables. To visualize Promoter Capture Hi-C data as heatmaps, rotated plaid plots were generated by the rotPlaid function supplied by the diffHic package on the merged dataset. Each chromosome was split in 1000 bins, and colored by the amount of reads in the interaction. Any interaction with more than 20 reads was colored a solid red.

**Overlapping enhancer regions with GWAS SNPs.** GWAS studies for T2D[6], BMI[8], and WHR[7] have identified 402, 941 and 463 distinct association signals, respectively. For IR we collected distinctive GWAS signals covering from studies of fasting insulin (FI) with and without adjustment for BMI[34,36,37], HOMA-IR[33], the modified Stumvoll Insulin Sensitivity Index (ISI)[38], and 53 genomic variants associated with both higher FI levels adjusted for BMI, lower HDL cholesterol levels and higher triglyceride levels[35], leading to a total of 82 distinct association signals with IR. Each of these four sets was expanded with SNPs in high LD (R2 > 0.8) with the original distinct association signals. Specifically, plink19 (http://www.cog-genomics. org/plink/1.9/)[77] was used to extract high LD SNPs within a 1 MB range of each SNP based on a subset (6148 Danish individuals) of the HRC imputed dataset used in the T2D GWAS[6]. The variant positions were converted into genome build38 before overlapping them with palmitate and TNFα responsive enhancer regions.

Regional plots were generated using standalone LocusZoom v1.4[78], as well as summary statistics available for T2D[6] and WHR[7].

**eQTL analysis.** The ADIGEN study participants[79,80] were selected from the Danish draft boards records. The study was approved by the Ethics Committee from the Capital Region of Denmark and informed consent was obtained from all participants in accordance with the Declaration of Helsinki II. Juvenile obesity was defined as weight 45% above the Metropolitan desirable weight (BMI ≥ 31 kg/m$^2$) at the draft board visit. 1930 obese individuals and 3601 randomly selected individuals for the population-representative control group were invited to participate in the study. In total 557 individuals volunteered to participate. From a subset of these Danish white men, 71 juvenile obese and 74 age-matched control individuals, skeletal muscle biopsies were taken under lidocaine local anesthesia from their right thigh using a thin Bergström needle and snap frozen in liquid nitrogen. The participants were healthy by self-report and under 65 years of age at the time of ADIGEN examination.

Gene expression analysis was performed by extracting total RNA using miRNeasy kit (Qiagen). The yield was optically measured and a randomly selected subset of the RNA samples were examined using an Experion electrophoresis station (BioRad) for integrity (RIN value), which was good in all cases. Gene expression of ~47,000 transcripts was measured by the HumanExpression HT-12 Chip (Illumina, USA). cRNA was synthesized from total RNA using the Nano Labeling Kit from Illumina (Epicentre), and the cRNA concentration was measured by Qubit fluorescent dye (Invitrogen, Germany) before loading the arrays. Hybridization was performed as recommended by Illumina and the Illumina HiScan was used to obtain the raw probe intensity level data. For failed expression arrays cRNA was resynthesized and rerun. The raw probe intensity values were exported from GenomeStudio without background correction and imported into R where the lumi package[81] was used for pre-processing. The array pre-processing included; quantile normalization, log2 transformation and probe filtering to remove probes with a detected P-value above 0.01.

The participants were genotyped using the Illumina CoreExome Chip v1.0 containing 538,448 genetic variants of which more than 240,000 are common. Genotypes were called using the Genotyping module (version 1.9.4) of GenomeStudio software (version 2011.1, Illumina) and Illumina HumanCoreExome-12v1-0_B.egt cluster file. The genotype data were subjected to standard quality control and then phased with EAGLE2[82] and imputed with the 1000 Genomes Project Phase III panel using Minimac3[83]. We selected 29 or 420 SNPs located within 11 or 124 enhancer regions, which changed activity by palmitate or TNFα treatment, respectively (see text for further description on how SNPs were selected). Only SNPs that were missing in less than 10% of the individuals, with an imputation quality (R2) higher than 0.4 and no significant deviation from Hardy-Weinberg equilibrium were extracted.

Matrix eQTL[84] was used to assess the association between 461 (TNFα) and 39 (palmitate) gene-SNP pairs (selected based on our Promoter Capture Hi-C data) in a total of 140 individuals with both expression and SNP data available (R version 3.5.0). To account for complex non-genetic factors, we used probabilistic estimation of expression residuals (PEER)[85]. Specifically, eQTL analysis was performed on inverse normal-transformed expression residuals adjusted for age, BMI-group (obese or control) and 15 PEER factors which is the number of factors recommended by the GTEX consortium[86] for studies with less than 150 individuals. The models were also run without the adjustment for BMI. Significant e-genes were identified after hierarchical multiple testing correction of the p-values

from TNFα and palmitate eQTL tests using the Bonferroni-BH procedure recommended by Huang et al.[87].

**Correlation analysis in BXD mice strains**. We selected and extracted the mean values of 48 metabolic phenotypes (Supplementary Data 7) that were measured across 42 and 37 BXD cohorts fed on CD and HFD, respectively[39–41,88]. Moreover, we extracted gene expression values of *Cep68*, *Gab2*, *Lamb1*, *Macf1*, *Eif6*, *Pabpc4*, *Btbd1*, *Filip1l*, *Tcea3*, *Nrp1*, *Zhx3*, *Tbx15*, and *Tnfaip8* from skeletal muscle tissue (quadriceps) (GSE60151)[89–91], adipose tissue (GN779; accessible on http://www.genenetwork.org/) and liver tissue (GSE60149)[40,41,91] from the different BXD mice strains. Spearmans rank correlation analysis was performed to determine significant associations between phenotypes and gene expression. The p-values from the 48 correlations from each diet and tissue were adjusted using false discovery rate correction (FDR)[92].

**Reporting summary**. Further information on research design is available in the Nature Research Reporting Summary linked to this article.

## Data availability

For analysis of RNA-seq data, reads were aligned to the hg38 GENCODE Comprehensive gene annotations [https://www.gencodegenes.org/]. H3K4me3 ChIP-seq peaks from human skeletal muscle myotubes were downloaded from Roadmap Epigenomics (sample E121) [https://egg2.wustl.edu/roadmap/data/byFileType/peaks/consolidated/narrowPeak/E121-H3K4me3.narrowPeak.gz]. Gene expression values from BXD cohorts were downloaded from the GEO data base entry GSE60151 and GSE60149, or from http://www.genenetwork.org/ [http://gn1.genenetwork.org/webqtl/main.py?FormID=sharinginfo&GN_AccessionId=779]. All novel sequencing data have been deposited in the NCBI Gene Expression Omnibus (GEO) and are accessible through GEO SuperSeries accession number GSE126102. RNA-seq data from GSE126101 have been used to generate Fig. 1a–i, Fig. 2j, k, Fig. 3e–h, Fig. 6c, and Supplementary Fig. 2. ChIP-seq data from GSE126099 have been used to generate Fig. 2a–i, Fig. 3e–h, Fig. 6b, Supplementary Fig. 3, and Supplementary Fig. 4. Promoter Capture Hi-C data from GSE126100 have been used to generate Fig. 3a–h, Fig. 6a, Supplementary Fig. 6, and Supplementary Fig. 7. The source data underlying Figs. 1H, I, 2G–J, 6B–E, 6I–M and Supplementary Fig. 1B–E, 4A–B, 4D, 5, 8A–D, and 9A–F are provided as a Source Data file.

## Code availability

All custom computer codes used for sequencing data analysis and figure generation are available at https://github.com/lars-work-sund/Skeletal-muscle-enhancer-interactions-identify-genes-controlling-human-metabolism [https://github.com/lars-work-sund/Skeletal-muscle-enhancer-interactions-identify-genes-controlling-human-metabolism].

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

## Acknowledgements

We thank Professor Peter Fraser for providing us with details about the Promoter Capture Hi-C protocol. We thank Professor Thorkild IA Sørensen for providing us with post-hoc access to data from the ADIGEN study. The Novo Nordisk Foundation Center for Basic Metabolic Research (http://www.metabol.ku.dk) is an independent research Center at the University of Copenhagen, partially funded by an unrestricted donation from the Novo Nordisk Foundation. This work was supported by an individual postdoc grant from the Independent Research Fund Denmark (DFF).

## Author contributions

K.W. and R.B. planned the study, collected all data, and wrote the first draft of the manuscript. K.W. performed RNA-seq, ChIP-seq, ChIP-qPCR, luciferase assays, and Promoter Capture HiC experiments. L.R.I. and C.T.W. conducted all bioinformatic analyses regarding ChIP-seq, RNA-seq, and Promoter Capture HiC data. J.B.J., N.G., and T.H. downloaded positions of IR, T2D, BMI, and WHR GWAS SNPs and overlapped with enhancer positions. J.B.J., R.R.M., A.A., O.P., N.G., and T.H. collected and performed experiments on human skeletal muscle biopsies. J.B.J, N.G., and T.H. performed eQTL analyses. M.W. and J.A. collected data from BXD mouse strains and performed correlation analyses. A.N.H. and L.S. assayed mitochondrial function, glucose uptake, glycogen synthesis, and AKT phosphorylation after knockdown of *Eif6*. All authors read and provided input to the final version of the paper.

## Competing interests

The authors declare no competing interests.

**Additional information**

