## [Peer Review File · Nature Communications]

Reviewers' comments:

Reviewer #1 (Remarks to the Author):

The authors perform ChIP-seq using two different marks, promoter capture HiC, and gene expression and genotyping arrays to identify enhancers in skeletal muscle, link enhancers to their targets, and identify GWAS variants which putatively act through insulin resistant specific enhancers on distal genes. Overall, the data appears strong, the experiments are designed well, and the results are interesting. Notably, the authors use promoter capture Hi-C and integrate in H3K27ac data, rather than Hi-ChIP, which is an excellent decision to prevent the confounding of H3K27ac allelic differences and loop differences. However, the manuscript is missing rigorous statistical analyses in several places, is poorly written, and has extremely sparse computational methods. Finally, as the paper does not show general novel biological principles, but instead provides novel data for identifying functional variation in a new condition/tissue type (which is a completely fine direction to take the paper for novelty), the authors need to make the data into a usable resource in order for it to be of value to the community. Specifically, the Hi-C data was not listed as a supplemental table – these and other data should be provided both as tables, and in bioinformatically ready formats (.bed for peaks, and .pjl, .juicebox, or .bedpe for loops).

Specific concerns for each section of the paper are separated below. In addition to these specific comments, the authors are advised to separate for each assay type the wet-lab and dry-lab methods into two sections, and to substantially expand upon the dry-lab methods (ie include versions of software used, precise statistical information, and how thresholds were chosen if they aren't 0.05 since these thresholds vary greatly throughout the paper). Additionally, many paragraphs and analyses could use introductions indicating why they are performed, and all analyses should have local interpretations within the paragraphs.

Major:

- Section 1 (gene expression):
 - o No concerns, as the analyses aptly demonstrated that genes are differentially expressed under treatments, and this expression is consistent with IR.
- Section 2 (ChIP-seqs):
 - o The authors should call broad peaks instead of narrow peaks due to using histone modification ChIP data, rather than TF binding ChIP data. The MACS2 manual and github discusses this difference and use case.
 - o The methods state that peaks were called with a p-value cutoff of 0.05, which will return a large number of false positive peak calls given multiple-testing burden. If this is a typo and it is a q-value cutoff of 0.05, this suggestion is minor instead of major and can be easily fixed.
 - o While the authors utilize H3K27ac and H3K4me1 to identify enhancers, they assume all H3K27ac and H3K4me1 to occur at enhancers. Other work, such as ROADMAP chromatin states, have shown H3K4me1 and H3K27ac to occur at promoter flanking chromatin and transcribed regions as well. The authors should do one of the following (or something similar) to ensure they are examining enhancers and not other regions: 1) perform H3K4me3 ChIP-seq in the same conditions; 2) use public H3K4me3 ChIP-seq from skeletal muscle in basal conditions to filter out regions not overlapping these peaks; or 3) remove putative enhancers near genes. Option 1 would be the cleanest, but it is understandable if the authors not wanting to generate more data (nor is it necessary given the quality of the current data). Option 3 would be the least clean, as it could easily filter out local enhancers.
- Section 3 (Hi-C):
 - o It is difficult to critique the statistics of this section, the number/quality of the loop calls, and suggest analysis that is possible with the current data due to the sparse methods, lack of statistical information for these analyses, and no reported sequencing depth.
 - o It is difficult to tell if Figure 3A is a large number of linked promoter enhancers, or if it is what would be expected due to the selected statistical significance criteria.
 - o While the authors found no significant differences between looping under Palmitate or TNFalpha

and the control cells, recent work by Greenwald et. al 2019 has shown that smaller fold changes are functional. From context, it appears that the authors did not observe these effects due to their sequencing depth; they should shift their conclusion away from a biological inference that chromatin conformation doesn't change in IR, to a conclusion about their statistical power.

o It is known that promoters with more enhancer connectivity are more cell type specific, and that integrating expression, enhancer, and conformation data can elucidate gene targets. The authors should rephrase their conclusions to be in support of these known biological relationships, rather than suggesting novelty. See <https://www.ncbi.nlm.nih.gov/pubmed/24213634>; <https://www.biorxiv.org/content/10.1101/299388v1>.

- Section 4 (Integrating GWAS):

o While the identification of the input variants for the eQTL analysis used some statistical information, there have been numerous recent papers using similar data to find causal GWAS variants; these manuscripts use fine-mapping to identify putative causal variants from GWAS by leveraging genomic annotations (for example <https://www.ncbi.nlm.nih.gov/pmc/articles/PMC3980523/>; <https://www.nature.com/articles/nature18642>; <https://www.biorxiv.org/content/10.1101/299388v1>). The paper would be greatly strengthened by fine-mapping the GWAS with either, or both, the enhancer and conformation data, and using these causal variants for eQTL testing.

o The conclusion of the paragraph on pg11 does not follow the data. The data does not describe chromatin interactions, it describes the association of these genes with T2D associated phenotypes in mouse.

o There are no methods to describe the statistics for Figure 5, other than a statement of "FDR correction" and "Spearman's correlation". Are the correlations shown on the scatter plots from Spearman? The plotted data in the scatter plots are values, not ranks. Were all possible pairs of phenotypes, tissue types, and genes tested for correlation for the heatmaps (it would seem so from Supp Figure 8)? If so, multiple testing should be done across all tests together, and a single FDR threshold should be selected beforehand.

o The authors utilize a p threshold of 0.05 for their eQTL section, and a q threshold of 0.2. As 0.2 is non-standard, it appears that it is the equivalent q-value to a p-value of 0.05 for their data. This is not an appropriate way to perform multiple test correction. A threshold needs to be selected before performing analysis, and the results need to be compared to this threshold.

o It appears that the authors test multiple SNPs against the same Gene during their eQTL analysis (as they use GWAS lead variants and variants in LD). If this is the case, they should not multiple test correct all variants for all genes simultaneously; instead, they should follow the process by most eQTL papers (namely GTEX) in which an empirical p-value is obtained from permutation testing within each gene, and the lead p-values from all eQTLs within each gene are then FDR corrected across all genes to identify significant eGenes.

o . The eQTL analysis should be repeated using PEER factors on gene expression as covariates, rather than PCs, using the appropriate number of PEER factors as suggested by GTEX; it is difficult from the methods to determine how many samples were used for the eQTL, but the suggested number of PEERs can be found at <https://github.com/broadinstitute/gtex-pipeline/tree/master/ctl.#2-calculate-peer-factors>

o It is unclear if any example eQTLs in Figures 6 and 7 are significant after multiple testing correction as their p-values are reported, rather than their q-values.

Minor:

- Section 1:

o The axes titles on Figure 1A are not clear or explained in the legend. What is a leading logFC?

o In general, it is difficult to know what figure corresponds with what sentence. Moving figure references to immediately after figure interpretation could help. Ie change "In total, we detected expression of 14,402 genes in skeletal muscle cells, of which 1,542 were regulated by palmitate treatment (621 downregulated and 921 upregulated) and 4,522 were changed by TNF α treatment (2,247 downregulated and 2,275 upregulated) (Figure 1B and 1C, Table S1, and Figure S2A)" to "In total, we detected expression of 14,402 genes in skeletal muscle cells, of which 1,542 were

regulated by palmitate treatment (621 downregulated and 921 upregulated; Figure 1B) and 4,522 were changed by TNF α treatment (2,247 downregulated and 2,275 upregulated; Figure 1C) (Table S1, and Figure S2A)”

o The GO analysis reads as though GO terms relevant for IR were arbitrarily chosen, and then differential expression for member genes were examined; from the tables, it seems that GO terms were identified from the differential expression, and then enrichments are plotted to help with visualization. Rewording this section would be helpful.

- Section 2:

o The authors use “enrichment” and “peaks” interchangeably regarding Figure 2A-C; it would be better to use “peaks” and/or “peak calls” to discuss these regions and their similarity.

o On line 109-113, “we found that most...H3K27ac (Figure 2A)” is written unclearly. It appears as though they are discussing how many peaks overlapped by any amount (which is fine if so); methods and results should be updated to clearly reflect the analysis performed

- Section 3:

o (Promoter) Capture Hi-C is sometimes referred to as Promoter Capture HiC, and sometimes only as Capture Hi-C. The authors should make this consistent, and use the clearer term Promoter Capture Hi-C

o The centromeres are not missing from HiC data because they don't form loops, it's because they are blacklisted due to mappability. The authors should remove this inference.

o While Figure 4B and 4C are great analyses and results make sense, their methods are difficult to understand

- Section 4:

o All *, **, and ****s should be reported in a visual legend, as well as the written legend for figures.

Reviewer #2 (Remarks to the Author):

This paper combines analysis of the transcriptome, enhancerome and chromatin interaction data of palmitate- or TNF α -treated human muscle cells treated with GWAS studies on BMI, insulin resistance and type 2 diabetes. A number of candidate gene are identified that may be involved in insulin resistance, which was further studied using the mouse BXD database. The paper is well written and the results are presented in a clear fashion. The main weakness of the paper is the lack of verification that any of the candidate genes influence IR. Without this evidence, the paper is unfinished. Basically, proof of the pudding that the chosen approach gives something meaningful is lacking. In addition, the paper adheres to a very unspecific interpretation of the concept of insulin resistance, leading to an overall very crude analysis that is expected to mostly yield false positives.

Main comments

1)What is missing from the manuscript is the demonstration that overexpression or deficiency of one or more of the candidate genes influences IR in cultured muscle cells or in vivo. In the present state, the manuscript follows a particular pipeline without providing evidence of a direct link with IR. The authors are strongly encouraged to explore whether inactivation of one or more of the candidate genes (via siRNA or CRISPR) influences the sensitivity of cultured muscle cells to insulin by measuring Akt phosphorylation or via any other relevant methodology.

2) I have reservations about the cell system used to model IR in muscle cells. Most of the transcriptomic and epigenetic changes induced by palmitate or TNF α will be completely disconnected from IR. As could be expected from the literature, palmitate and TNF α treatment lead to major changes in gene expression in muscle cells, with some overlap. This system seems too crude and unspecific to permit the identification of potential candidate genes participating in metabolic dysfunction. It is expected that only a tiny minority of genes induced by palmitate or

TNF α have any relevance for IR. Accordingly, the claim that the results identify the enhancerome of IR in human skeletal muscle cells is inappropriate or even misleading.

3) If the focus of the paper is on IR, why were GWAS SNPs associated with T2DM, BMI or waist-to-hip ratio included as well. It seems more fitting that the analysis is limited to SNPs associated with IR. The far majority of the SNPs associated with T2DM, BMI or waist-to-hip ratio are not associated with IR but with other features (beta cell function, regulation of satiety, etc)

Additional comments

1) Line 140-142: This sentence is misleading. The enhancerome of TNF α and palmitate was identified, not the enhancerome of IR. Moreover, support a role of what?

2) Page 7. What is the basis for the statement: "Enhancers with the strongest increase in H3K27ac after palmitate treatment were located at the proximity of genes related to fatty acid metabolism...". Was any quantitative analysis done to support this statement. Merely mentioning two examples (PDK4 and ANGPTL4) is not sufficient. Same for the contention that enhancers strongly regulated by TNF α included elements located close to cytokine genes.

3) The analysis of the enhancer-promoter interactions related to FTO intron 1 is out of context and distracting (Figure 3D). The data should be removed or should at least moved to the supplemental data.

4) Figure 1 and 2 contain an interesting analysis of the transcriptomic and epigenomic effects of palmitate and TNF α in human muscle cells, while figure 3 present the enhancer-promoter interactions triggered by palmitate and TNF α . However, I don't see how these data bear any relevance to IR. The paper makes an inappropriate and unsuccessful attempt to frame the results in the context of IR.

5) It is unclear why the paper broadly describes the analysis of the enhancer-promoter interactions in human muscle cells, given that neither palmitate nor TNF α caused any changes in mapped ChIP interactions.

6) Line 165-166. This statement is an over-extrapolation of the actual data. Rather, the results suggest that chromatin conformation doesn't respond to palmitate or TNF α . Moreover, the fact that neither palmitate nor TNF α treatment, despite causing massive changes at the transcriptome and enhancerome level, have any effect on mapped ChIP interactions seems rather unlikely, thereby questioning the validity of the ChIP data.

7) The description of the data from line 168 to 176 is of no relevance to the paper, as is the case for the entire figure 3.

8) It is unclear how the results described in lines 179 and 192 can lead to the statement in line 192-194.

Response to reviewers

Reviewer #1:

The authors perform ChIP-seq using two different marks, promoter capture HiC, and gene expression and genotyping arrays to identify enhancers in skeletal muscle, link enhancers to their targets, and identify GWAS variants which putatively act through insulin resistant specific enhancers on distal genes. Overall, the data appears strong, the experiments are designed well, and the results are interesting. Notably, the authors use promoter capture Hi-C and integrate in H3K27ac data, rather than Hi-ChIP, which is an excellent decision to prevent the confounding of H3K27ac allelic differences and loop differences. However, the manuscript is missing rigorous statistical analyses in several places, is poorly written, and has extremely sparse computational methods. Finally, as the paper does not show general novel biological principles, but instead provides novel data for identifying functional variation in a new condition/tissue type (which is a completely fine direction to take the paper for novelty), the authors need to make the data into a usable resource in order for it to be of value to the community. Specifically, the Hi-C data was not listed as a supplemental table – these and other data should be provided both as tables, and in bioinformatically ready formats (.bed for peaks, and .pgl, .juicebox, or .bedpe for loops).

Response: We thank the reviewer for the positive comments and constructive criticism. We have addressed the reviewer's concerns and provide a point-by-point response below. We have highlighted our changes in red in the revised manuscript to enhance clarity.

We have listed the Hi-C data as a .bedpe file in Supplemental Table S4.

Specific concerns for each section of the paper are separated below. In addition to these specific comments, the authors are advised to separate for each assay type the wet-lab and dry-lab methods into two sections, and to substantially expand upon the dry-lab methods (ie include versions of software used, precise statistical information, and how thresholds were chosen if they aren't 0.05 since these thresholds vary greatly throughout the paper).

Additionally, many paragraphs and analyses could use introductions indicating why they are performed, and all analyses should have local interpretations within the paragraphs

Response: We have followed the reviewer's advice and made substantial re-writing of the methods section, where we now separate the wet-lab and dry-lab sections. Moreover, we have included more detailed introductions and conclusions to specific paragraphs. These changes are highlighted in red font in the revised manuscript.

Major:

- Section 1 (gene expression):
 - o No concerns, as the analyses aptly demonstrated that genes are differentially expressed under treatments, and this expression is consistent with IR.

Response: Thank you. No changes to the gene expression analysis was done.

- Section 2 (ChIP-seqs):

o The authors should call broad peaks instead of narrow peaks due to using histone modification ChIP data, rather than TF binding ChIP data. The MACS2 manual and github discusses this difference and use case.

Response: Thank you for this suggestion. After looking into Encode guidelines for analysis of histone modification ChIP-seq data, we agree with the reviewer that some histone modifications should be analyzed as broad peaks. Encode lists H3K4me1 as a broad peak, whereas H3K27ac is listed as a narrow peak (<https://www.encodeproject.org/chip-seq/histone/>). Therefore, we have re-analyzed the H3K4me1 ChIP-seq data using broad peak calling. This reduced the number of H3K4me1 peaks from 146,624 to 107,405 (see updated Figure 2A and supplemental Table S3).

o The methods state that peaks were called with a p-value cutoff of 0.05, which will return a large number of false positive peak calls given multiple-testing burden. If this is a typo and it is a q-value cutoff of 0.05, this suggestion is minor instead of major and can be easily fixed.

Response: This is a misunderstanding, and we have updated the methods section to more clearly describe what was done. The consensus peaks analyzed were defined using the IDR algorithm (*Li et al., Measuring reproducibility of high-throughput experiments, The Annals of Applied Statistics 2011*), as recommended by the ENCODE consortium. Prior to defining consensus peaks, samples were checked to ensure that the ChIP-step had worked. In addition to visually inspect the alignments, the number of peaks called was used as a metric. If the ChIP-step fails only few peaks can be called relative to samples where the enrichment succeeded, regardless of which cutoffs are used. We used raw P-values for testing for successful ChIP as recommended by the ENCODE 2012 pipeline. The P-value defined peaks were not used for any analysis, except as input to the IDR software.

o While the authors utilize H3K27ac and H3K4me1 to identify enhancers, they assume all H3K27ac and H3K4me1 to occur at enhancers. Other work, such as ROADMAP chromatin states, have shown H3K4me1 and H3K27ac to occur at promoter flanking chromatin and transcribed regions as well. The authors should do one of the following (or something similar) to ensure they are examining enhancers and not other regions: 1) perform H3K4me3 ChIP-seq in the same conditions; 2) use public H3K4me3 ChIP-seq from skeletal muscle in basal conditions to filter out regions not overlapping these peaks; or 3) remove putative enhancers near genes. Option 1 would be the cleanest, but it is understandable if the authors not wanting to generate more data (nor is it necessary given the quality of the current data). Option 3 would be the least clean, as it could easily filter out local enhancers.

Response: We agree with the reviewer that we also include some promoter regions when using the overlap of H3K4me1 and H3K27ac ChIP-seq peaks to identify enhancers. We therefore excluded all active promoters by downloading H3K4me3 ChIP-seq peaks from human skeletal muscle myotubes (derived from skeletal muscle myoblasts) from Roadmap Epigenomics (*Bernstein et al., The NIH Roadmap Epigenomics Mapping Consortium, Nature Biotechnology 2010*) (sample E121). The data was lifted to hg38 and filtered to keep only peaks with an FDR < 0.05. Active promoters were defined as RefSeq gene promoters with a H3K4me3 peak within 3000bp upstream to 1000bp downstream of its TSS. This analysis identified 25,465 active promoters. Next, we filtered out promoters from our enhancer mapping by defining enhancers as regions containing a consensus peak of both H3K27ac and H3K4me1 *and* that was more than 3000bp upstream or 1000bp downstream of the TSS of an active H3K4me3 covered promoter (see updated Figure 2A).

- Section 3 (Hi-C):

o It is difficult to critique the statistics of this section, the number/quality of the loop calls, and suggest analysis that is possible with the current data due to the sparse methods, lack of statistical information for these analyses, and no reported sequencing depth.

Response: We thank the reviewer for pointing at a lack of clarity. We agree with the reviewer that the methods describing the Promoter Capture Hi-C analysis could be more extensively described. We have re-written the section concerning the bioinformatic analysis of the promoter-capture Hi-C data.

Information on the sequencing depth is given in Supplemental Table S11.

o It is difficult to tell if Figure 3A is a large number of linked promoter enhancers, or if it is what would be expected due to the selected statistical significance criteria.

Response: The abovementioned adjustments of the enhancer mapping (re-analyzing H3K4me1 ChIP-seq data using broad peak detection and sorting out promoter regions) required that we also re-analyzed the Promoter Capture Hi-C data. This is because the Hi-C analysis was set up so that we only test for interactions between a captured promoter and a predefined enhancer region (based on our ChIP-seq analyses). Since the total number of enhancers decreased (mainly by removing promoter regions) the total number of interactions between a promoter and an enhancer also decreased from 39,098 to 36,809 (see updated Figure 3A). Other studies using promoter capture Hi-C reports between 70,000 and 300,000 promoter-interacting regions (*Choy et al., Promoter interactome of human embryonic stem cell-derived cardiomyocytes connects GWAS regions to cardiac gene networks. Nature Commun. 2018; Sahlén et al., Genome-wide mapping of promoter-anchored interactions with close to single-enhancer resolution. Genome Biology 2015; Siersbæk et al., Dynamic Rewiring of Promoter-Anchored Chromatin Loops during Adipocyte Differentiation. Mol Cell 2017*). However, these studies do not use a set of pre-defined enhancers but instead test for promoter interactions throughout the genome (also including promoter-promoter interactions). Thus, we believe that it is difficult to directly compare the number of linked promoter-enhancer interactions that we find to these studies. Another study using regular Hi-C, but only investigating promoter-enhancer interactions, identified 29,132 enhancer–promoter interactions involving 6,133 active promoters and 15,432 distal active enhancers (*Jin et al., A high-resolution map of the three-dimensional chromatin interactome in human cells. Nature 2013*), compared to our study where we identify 36,809 interactions involving 10,237 promoters and 31,994 enhancers.

o While the authors found no significant differences between looping under Palmitate or TNFalpha and the control cells, recent work by Greenwald et. al 2019 has shown that smaller fold changes are functional. From context, it appears that the authors did not observe these effects due to their sequencing depth; they should shift their conclusion away from a biological inference that chromatin conformation doesn't change in IR, to a conclusion about their statistical power.

Response: Thank you for this very good comment. We agree with the reviewer that we cannot conclude that the lack of dynamic chromatin looping upon treatment with palmitate or TNF α cannot be explained by a lack of power in the analysis. We have therefore re-written our conclusions about this section. We would like to point however, that our data resembles very well with a previous study (*Jin et al., A high-resolution map of the three-dimensional chromatin interactome in human cells. Nature 2013*), which characterized the dynamics of promoter–enhancer contacts after TNF α signaling in human fibroblasts. There TNF α -responsive enhancers were shown to be already in contact with their target promoters, before transient activation or repression of enhancer activity by

TNF α treatment. This observation is to be opposed to chromatin interactions at cell-type-specific enhancers (when comparing fibroblasts to embryonic stem cells), where chromatin interactions appear to be variable. The discrepancy between TNF α -dependent and cell type-specific enhancers was found to be correlated with the levels of H3K4me1 at the dynamically regulated enhancers. Thus, despite the quick induction of the H3K27ac mark at TNF α -responsive enhancers, the strength of H3K4me1 signal was largely unchanged. This is in good alignment with our findings, where we see large changes in H3K27ac at palmitate and TNF α -responsive enhancers, correlating with strong induction of gene transcription (Figure 2F-K), but we do not detect major changes in the H3K4me1 levels at these enhancers (Supplemental Figure S4B and C).

In line with this, most papers detecting dynamic chromatin interactions are studying cell-type-specific chromatin conformation (*Siersbæk et al., Dynamic Rewiring of Promoter-Anchored Chromatin Loops during Adipocyte Differentiation. Mol Cell 2017, Greenwald et al., Subtle changes in chromatin loop contact propensity are associated with differential gene regulation and expression. Nat Commun. 2019, Javierre et al., Lineage-Specific Genome Architecture Links Enhancers and Non-coding Disease Variants to Target Gene Promoters. Cell 2016*). And even in this context, a recent paper detecting dynamic promoter-enhancer contacts during epidermal differentiation, reports that the majority of differentiation-induced genes have pre-established chromatin contacts - again suggesting that the majority of gene activation is not associated with chromatin remodeling (*Rubin et al., Lineage-specific dynamic and pre-established enhancer–promoter contacts cooperate in terminal differentiation. Nature genetics 2017*).

o It is known that promoters with more enhancer connectivity are more cell type specific, and that integrating expression, enhancer, and conformation data can elucidate gene targets. The authors should rephrase their conclusions to be in support of these known biological relationships, rather than suggesting novelty.

See <https://www.ncbi.nlm.nih.gov/pubmed/24213634>; <https://www.biorxiv.org/content/10.1101/299388v1>.

Response: We decided to remove the GO analysis of the bottom promoters (<2 enhancer interactions) and top promoters (>12 promoters) since it was also commented by Reviewer #2 as irrelevant for the paper. Moreover, we have moved the ECDF plots from Figure 4 to Figure 3 so that they serve as a validation of our Promoter Capture Hi-C data, instead of suggesting a novel a relationship between expression, enhancer, and conformation data.

- Section 4 (Integrating GWAS):

- o While the identification of the input variants for the eQTL analysis used some statistical information, there have been numerous recent papers using similar data to find causal GWAS variants; these manuscripts use fine-mapping to identify putative causal variants from GWAS by leveraging genomic annotations (for example

- <https://www.ncbi.nlm.nih.gov/pmc/articles/PMC3980523/>; <https://www.nature.com/articles/nature18642>; <https://www.biorxiv.org/content/10.1101/299388v1>). The paper would be greatly strengthened by fine-mapping the GWAS with either, or both, the enhancer and conformation data, and using these causal variants for eQTL testing.

Response: We appreciate the suggestion given by the reviewer to use fine-mapping to identify causal GWAS variants before performing the eQTL analysis. However, we believe that the approach we use here is very similar and equally valid. In our study, we identify causal GWAS SNPs by using already defined significant SNPs (as well as SNPs in high LD to these) and from here select only those that

overlap enhancer regions that change activity by either palmitate or TNF α . Before performing the eQTL analysis, we further integrated the Promoter capture Hi-C data, as well as the gene expression data, so that we only selected those SNPs that were found in an enhancer region and linked to a gene promoter with a simultaneous change in gene expression (in the same direction as the enhancer activity). This approach is illustrated in Figure 4B in the manuscript.

o The conclusion of the paragraph on pg11 does not follow the data. The data does not describe chromatin interactions, it describes the association of these genes with T2D associated phenotypes in mouse.

Response: Thank you. This paragraph has been rewritten.

o There are no methods to describe the statistics for Figure 5, other than a statement of “FDR correction” and “Spearman correlation”. Are the correlations shown on the scatter plots from Spearman? The plotted data in the scatter plots are values, not ranks. Were all possible pairs of phenotypes, tissue types, and genes tested for correlation for the heatmaps (it would seem so from Supp Figure 8)? If so, multiple testing should be done across all tests together, and a single FDR threshold should be selected beforehand.

Response: The correlations were calculated from a Spearman correlation analysis, however, in the first version of the manuscript, the plotted data on the scatter plots were the actual data values with a linear regression line for visualization. We agree with the reviewer that it is more accurate to plot the ranked values, so we have changed this on all scatter plots correlating gene expression data and BXD phenotypes. Correction for multiple testing of the BXD correlations were done by correcting all correlations for each gene in each tissue when the mice were on either CD or HFD (see updated Table S8-S10). An FDR threshold of 0.2 was chosen (see below).

o The authors utilize a p threshold of 0.05 for their eQTL section, and a q threshold of 0.2. As 0.2 is non-standard, it appears that it is the equivalent q-value to a p-value of 0.05 for their data. This is not an appropriate way to perform multiple test correction. A threshold needs to be selected before performing analysis, and the results need to be compared to this threshold.

Response: We agree that using a FDR threshold of 0.2 is non-standard. However, for the eQTL analysis, we tested only gene-SNP pairs where the SNP was located in an enhancer that changed activity by either palmitate or TNF α *and* where the gene expression was changed in the same direction as the enhancer activity *and* where we also detected a significant Promoter Capture Hi-C interaction. Thus, already before performing the eQTL analysis, we have evidence of a connection between the SNP and the gene. We believe that many of the targets, that did not turn out significant in the eQTL analysis, could actually still be potentially regulated by their respective GWAS SNPs under conditions where the cells are exposed to fatty acids or inflammatory signaling. We used the eQTL, as well as the BXD expression correlation analysis, to further narrow down the list of potential targets and therefore decided to be less stringent in our FDR correction. In the manuscript, we highlight *TBX15*, which had only a weak association with WHR SNPs, but none the less has a strong biological link to regulation of skeletal muscle metabolism and fat mass distribution.

o It appears that the authors test multiple SNPs against the same Gene during their eQTL analysis (as they use GWAS lead variants and variants in LD). If this is the case, they should not multiple test correct all variants for all genes simultaneously; instead, they should follow the process by most

eQTL papers (namely GTEx) in which an empirical p-value is obtained from permutation testing within each gene, and the lead p-values from all eQTLs within each gene are then FDR corrected across all genes to identify significant eGenes.

Response: Thank you for this comment. For some genes we tested only a single genetic variant but for other genes we did indeed test several genetic variants. We have therefore followed the procedure suggested by the reviewer and have now performed hierarchical correction of our p-values. To identify significant eGenes, we used the Bonferroni-BH procedure which is recommended by Huang et al. (*Huang et al., Power, false discovery rate and Winner's Curse in eQTL studies. NAR, 2018*). This paper compares different multiple correction methods in eQTL studies, including the FastQTL permutation FDR correction used in GTEx.

o . The eQTL analysis should be repeated using PEER factors on gene expression as covariates, rather than PCs, using the appropriate number of PEER factors as suggested by GTEx; it is difficult from the methods to determine how many samples were used for the eQTL, but the suggested number of PEERs can be found at <https://github.com/broadinstitute/gtex-pipeline/tree/master/ctl.#2-calculate-peer-factors>

Response: We have followed this suggestion and have now performed the eQTL analysis where we adjust for 15 PEER factors rather than the PCs. Moreover, we have added the number of individuals used (n=140) in the eQTL analyses in the *eqtl-statistics method* section.

o It is unclear if any example eQTLs in Figures 6 and 7 are significant after multiple testing correction as their p-values are reported, rather than their q-values.

Response: In the original version of the manuscript we highlighted *TIMP4* as a potential target gene of T2D associated SNP rs11712037. However, after performing the eQTL analysis adjusting for PEER factors and performed hierarchical correction of our p-values, the *TIMP4*/ rs11712037 eQTL was no longer significant. The example given in the new Figure 6 (*EIF6*) is significant and we have now added the FDR-value for the eQTL test to the figure.

Minor:

• Section 1:

o The axes titles on Figure 1A are not clear or explained in the legend. What is a leading logFC?

Response: Methods and Figure legends have been updated to explain the leading logFC.

o In general, it is difficult to know what figure corresponds with what sentence. Moving figure references to immediately after figure interpretation could help. Ie change “In total, we detected expression of 14,402 genes in skeletal muscle cells, of which 1,542 were regulated by palmitate treatment (621 downregulated and 921 upregulated) and 4,522 were changed by TNF α treatment (2,247 downregulated and 2,275 upregulated) (Figure 1B and 1C, Table S1, and Figure S2A)” to “In total, we detected expression of 14,402 genes in skeletal muscle cells, of which 1,542 were regulated by palmitate treatment (621 downregulated and 921 upregulated; Figure 1B) and 4,522 were changed by TNF α treatment (2,247 downregulated and 2,275 upregulated; Figure 1C) (Table S1, and Figure S2A)”

Response: Thank you for this suggestion. We have made every effort to follow this advice throughout the manuscript.

o The GO analysis reads as though GO terms relevant for IR were arbitrarily chosen, and then differential expression for member genes were examined; from the tables, it seems that GO terms were identified from the differential expression, and then enrichments are plotted to help with visualization. Rewording this section would be helpful.

Response: To represent the GO analysis in an unbiased way, we decided to plot the top 10 up or down regulated GO terms from the analyses of palmitate and TNF α regulated genes (see Figure 1D-G).

• Section 2:

o The authors use “enrichment” and “peaks” interchangeably regarding Figure 2A-C; it would be better to use “peaks” and/or “peak calls” to discuss these regions and their similarity.

Response: We have revised the manuscript accordingly.

o On line 109-113, “we found that most...H3K27ac (Figure 2A)” is written unclearly. It appears as though they are discussing how many peaks overlapped by any amount (which is fine if so); methods and results should be updated to clearly reflect the analysis performed.

Response: We have added a sentence to the ChIP-seq methods highlighting that any amount of overlap was considered an overlap.

• Section 3:

o (Promoter) Capture Hi-C is sometimes referred to as Promoter Capture HiC, and sometimes only as Capture Hi-C. The authors should make this consistent, and use the clearer term Promoter Capture Hi-C

Response: We have corrected this and used the term *Promoter Capture Hi-C* throughout the manuscript.

o The centromeres are not missing from HiC data because they don't form loops, it's because they are blacklisted due to mappability. The authors should remove this inference.

Response: We have removed this statement.

o While Figure 4B and 4C are great analyses and results make sense, their methods are difficult to understand

Response: We have moved the plots from Figure 4B and C to Figure 3E-J since we think they serve as a nice validation of our Promoter Capture Hi-C data. The plots represent standard Empirical Cumulative Distribution function (EDCF) plots, where the X-axis is the RNA-seq logFC and the y-axis is the fraction of genes with this logFC or less.

• Section 4:

o All *, **, and ***s should be reported in a visual legend, as well as the written legend for figures.

Response: We appreciate the suggestion given by the reviewer, however as the figure panels in the manuscript are already quite compact, we decided not to include an extra legend explaining *, **, and ***, since we do not consider this as standard procedure. The meaning of *, **, and *** is clearly described in the figure legends.

Reviewer #2:

This paper combines analysis of the transcriptome, enhancerome and chromatin interaction data of palmitate- or TNF α -treated human muscle cells treated with GWAS studies on BMI, insulin resistance and type 2 diabetes. A number of candidate gene are identified that may be involved in insulin resistance, which was further studied using the mouse BXD database. The paper is well written and the results are presented in a clear fashion. The main weakness of the paper is the lack of verification that any of the candidate genes influence IR. Without this evidence, the paper is unfinished. Basically, proof of the pudding that the chosen approach gives something meaningful is lacking. In addition, the paper adheres to a very unspecific interpretation of the concept of insulin resistance, leading to an overall very crude analysis that is expected to mostly yield false positives.

We thank the reviewer for the positive comments on our work and on the manuscript. The reviewer raises a very valid point about the physiological relevance of our results. We have performed additional experiments to show the functional relevance of one of the genes we identified through our pipeline. These results are now presented in the revised manuscript. We provide a point-by-point response to the reviewer's comments and we have highlighted our changes in red in the revised manuscript to enhance clarity.

Main comments

1) What is missing from the manuscript is the demonstration that overexpression or deficiency of one or more of the candidate genes influences IR in cultured muscle cells or in vivo. In the present state, the manuscript follows a particular pipeline without providing evidence of a direct link with IR. The authors are strongly encouraged to explore whether inactivation of one or more of the candidate genes (via siRNA or CRISPR) influences the sensitivity of cultured muscle cells to insulin by measuring Akt phosphorylation or via any other relevant methodology.

Response: We thank the reviewer for this important comment. As written in more detail below, we agree with the reviewer that it was an overstatement to suggest that palmitate and TNF α treated cultured myotubes serve as a good model of skeletal muscle insulin resistance. The intention of the paper was not to study the mechanisms of IR, but instead induce metabolic stress (lipotoxicity and inflammation) to muscle cells, which are both treatments that have been associated with insulin resistance and the metabolic syndrome. By exposing muscle cells to these treatments, we were able to follow concurrent changes in enhancer activity and connected promoter transcription in order to find novel targets of metabolic GWAS SNPs that are overlapping our enhancer regions. Thus, we identify candidate genes involved in the regulation of IR, T2D, WHR or BMI.

In regards to the validation of our candidate list, we have provided further experiments validating our findings. We have extensively characterized one of the targets, *EIF6*, which we find linked to the regulation WHR and BMI in humans, and to exercise performance in mice, (Figure 6I-M, Figure S8 and Figure S9). Our new experiments demonstrate that siRNA-mediated knockdown of *Eif6* in

murine muscle cells causes a lower mitochondrial respiration and protein levels of mitochondrial complex II, whereas we do not detect any changes in insulin-stimulated glucose uptake, glycogen synthesis, or AKT phosphorylation. These findings are in agreement with our finding that *Eif6*-linked SNPs associate with WHR and BMI, but not with IR or T2D. Moreover, we find that *EIF6* expression is downregulated by TNF α treatment, which corresponds to findings demonstrating that TNF α reduces mitochondrial function in muscle cells (*McLean et al., Tumor necrosis factor- α (TNF) effects on mitochondrial metabolism in C2C12 myotubes. Physiology 2013. Dun et al., Low molecular weight guluronate prevents TNF- α -induced oxidative damage and mitochondrial dysfunction in C2C12 skeletal muscle cells. Food Funct. 2015*). Thus, we speculate that the effect of TNF α exposure on mitochondrial function could be partly mediated by regulation of *Eif6* levels.

2) I have reservations about the cell system used to model IR in muscle cells. Most of the transcriptomic and epigenetic changes induced by palmitate or TNF α will be completely disconnected from IR. As could be expected from the literature, palmitate and TNF α treatment lead to major changes in gene expression in muscle cells, with some overlap. This system seems too crude and unspecific to permit the identification of potential candidate genes participating in metabolic dysfunction. It is expected that only a tiny minority of genes induced by palmitate or TNF α have any relevance for IR. Accordingly, the claim that the results identify the enhancerome of IR in human skeletal muscle cells is inappropriate or even misleading.

Response: While we agree with the reviewer that palmitate and TNF α treatment in cultured myotubes does not represent a good model of skeletal muscle insulin resistance, we hope the reviewer will appreciate that the real advantage of this cell culture system is that we induce two separate stresses to the muscle cells (by palmitate or TNF α treatment) that are relevant to metabolic disease and which cause massive changes to the activity of enhancers and genes. By overlapping the enhancer regions with GWAS SNP of metabolic diseases, and information of chromatin conformation, we were able to concurrently follow changes in the activity of enhancers encompassing GWAS SNPs and transcription from a connected promoter – thereby establishing links between GWAS SNPs and gene targets (which is the main purpose of the paper). We have re-written the manuscript to clarify this and removed all statements related to “insulin resistance” and replaced by a more conservative description, now mentioning lipid toxicity and response to proinflammatory cytokine.

3) If the focus of the paper is on IR, why were GWAS SNPs associated with T2DM, BMI or waist-to-hip ratio included as well. It seems more fitting that the analysis is limited to SNPs associated with IR. The far majority of the SNPs associated with T2DM, BMI or waist-to-hip ratio are not associated with IR but with other features (beta cell function, regulation of satiety, etc).

Response: As written above, the intention of the paper was not to study the mechanisms of IR, but instead induce a stress to muscle cells resembling stresses encountered within metabolic disease, which would enable us to simultaneously follow changes in enhancer activity and connected promoter transcription, in order to find novel targets of metabolic GWAS SNPs. As circulating free fatty acids and proinflammatory cytokines represent an important link between obesity, insulin resistance, and T2D, we decided to include GWAS of all these diseases. We are aware that this was not clear from reading the manuscript and we have re-written the introduction to clarify this.

Additional comments

1) Line 140-142: This sentence is misleading. The enhancerome of TNF α and palmitate was identified, not the enhancerome of IR. Moreover, support a role of what?

Response: We agree with the reviewer. We have rephrased this sentence.

2) Page 7. What is the basis for the statement: “Enhancers with the strongest increase in H3K27ac after palmitate treatment were located at the proximity of genes related to fatty acid metabolism...”. Was any quantitative analysis done to support this statement. Merely mentioning two examples (PDK4 and ANGPTL4) is not sufficient. Same for the contention that enhancers strongly regulated by TNF α included elements located close to cytokine genes.

Response: Thank you for this comment. We did not mean to make a generalized statement about the enhancers that are upregulated by palmitate or TNF α treatment, but we meant to show examples of enhancers that are strongly regulated by palmitate and TNF α . For example, the *PDK4*-10 kb enhancer, which is the strongest upregulated enhancer by palmitate, represents a nice example since we also find *PDK4* expression to be upregulated by palmitate exposure, which agrees with previous findings (for example: *Huang et al., Regulation of Pyruvate Dehydrogenase Kinase Expression by Peroxisome Proliferator-Activated Receptor- α Ligands, Glucocorticoids, and Insulin. Diabetes 2002*). We have re-written this paragraph to make it clearer.

3) The analysis of the enhancer-promoter interactions related to FTO intron 1 is out of context and distracting (Figure 3D). The data should be removed or should at least moved to the supplemental data.

Response: Thank you for this comment. We agree with the reviewer that reporting enhancer-promoter interactions related to FTO was distracting and thus, we have removed this figure panel from the manuscript.

4) Figure 1 and 2 contain an interesting analysis of the transcriptomic and epigenomic effects of palmitate and TNF α in human muscle cells, while figure 3 present the enhancer-promoter interactions triggered by palmitate and TNF α . However, I don't see how these data bear any relevance to IR. The paper makes an inappropriate and unsuccessful attempt to frame the results in the context of IR.

Response: Thank you the positive comment. As discussed above, we agree with the reviewer that it was an overstatement to suggest that palmitate and TNF α treated cultured myotubes serve as a good model of skeletal muscle insulin resistance. The intention of the paper was not to study the mechanisms of IR, but instead induce metabolic stress to muscle cells, which would enable us to follow concurrent changes in enhancer activity and connected promoter transcription in order to find novel targets of metabolic GWAS SNPs.

5) It is unclear why the paper broadly describes the analysis of the enhancer-promoter interactions in human muscle cells, given that neither palmitate nor TNF α caused any changes in mapped cHiC interactions.

Response: We apologize for the lack of clarity. We performed Promoter Capture Hi-C in all conditions (ctrl, palmitate or TNF α treated myotubes) in order to investigate if any of the detected enhancer activations correlated with a dynamic chromatin looping. However, we did not find this to

be the case. This agrees with a previous study (Jin *et al.*, *A high-resolution map of the three-dimensional chromatin interactome in human cells. Nature 2013*), where the dynamics of promoter–enhancer contacts after TNF α treatment in human fibroblasts was characterized. In this study, TNF α -responsive enhancers are already in contact with their target promoters before transient activation or repression of enhancer activity by TNF α treatment. Thus, despite the quick induction of the H3K27ac mark at TNF α -responsive enhancers, the strength of the H3K4me1 signal, as well as the chromatin conformation, was largely unchanged. This is consistent with our observations, where we detected large changes in H3K27ac at palmitate and TNF α -responsive enhancers, correlating with strong induction of gene transcription (Figure 2F-K), but did not detect major changes in the H3K4me1 levels at these enhancers (Supplemental Figure S4B and C).

Instead, we analyzed the general chromatin conformation of myotubes, which to our knowledge is the first Hi-C data set of human myotubes. We believe our Promoter Capture Hi-C dataset is still relevant to the manuscript, as it gives us information on which enhancers are connected to a given promoter in myotubes, and thus, can help elucidate which gene are regulated by a given enhancer.

6) Line 165-166. This statement is an over-extrapolation of the actual data. Rather, the results suggest that chromatin conformation doesn't respond to palmitate or TNF α . Moreover, the fact that neither palmitate nor TNF α treatment, despite causing massive changes at the transcriptome and enhancerome level, have any effect on mapped cHiC interactions seems rather unlikely, thereby questioning the validity of the cHiC data.

Response: We agree with the reviewer that we cannot rule out that our observation of stable chromatin conformation under palmitate or TNF α treatment was not due to a lack of power in our analysis. Thus, we have re-written this paragraph. However, as mentioned above, our data aligns with a previous study (Jin *et al.*, *A high-resolution map of the three-dimensional chromatin interactome in human cells. Nature 2013*) where dynamics of promoter–enhancer contacts after TNF α signaling in human fibroblasts show that TNF α -responsive enhancers are already in contact with their target promoters - despite a massive induction of the H3K27ac mark at TNF α -responsive enhancers and in gene transcription.

Another recent paper, detecting dynamic promoter-enhancer contacts during epidermal differentiation, also finds that the majority of differentiation-induced genes have pre-established chromatin contacts - again suggesting that the majority of gene activation is not associated with chromatin remodeling (Rubin *et al.*, *Lineage-specific dynamic and pre-established enhancer–promoter contacts cooperate in terminal differentiation. Nature genetics 2017*).

7) The description of the data from line 168 to 176 is of no relevance to the paper, as is the case for the entire figure 3.

Response: We decided to remove the GO analysis of the bottom promoters (<2 enhancer interactions) and top promoters (>12 promoters) since it was also commented by Reviewer#1 as irrelevant for the paper. We still think the Promoter Capture Hi-C data (Figure 3) is relevant to the manuscript, as it gives us information on which enhancers are connected to a given promoter in myotubes, and thus, can help elucidate which gene are regulated by a given enhancer.

8) It is unclear how the results described in lines 179 and 192 can lead to the statement in line 192-194.

Response: We have moved the ECDF plots from Figure 4 to Figure 3E-J since we think they serve as a nice validation of our Promoter Capture Hi-C data. We have also re-written the paragraph in the result section describing these plots in order to clarify.

Reviewers' comments:

Reviewer #1 (Remarks to the Author):

The authors have satisfactorily addressed all my comments.

Kelly A Frazer

Reviewer #3 (Remarks to the Author):

In this revised manuscript, Williams and colleagues report results of detailed epigenetic analyses of human myotubes treated with palmitate or TNF α . Integrated analysis of RNA-seq, ChIP seq, and Hi-C together with GWAS data for insulin resistance-related traits results in identification of multiple loci potentially linked to metabolic disease and muscle responses. These data are then used to query mouse tissue expression data to identify genes also correlated with key metabolic traits in mice.

The manuscript is well written and presented clearly. The authors have been responsive to the prior review, adding analysis of mitochondrial oxidative metabolism in cells with knockdown of Eif6. Overall enthusiasm is reduced by the authors' approach to overlap with GWAS, as this may reduce the ability to conclude about transcriptional regulatory effects imposed by the acute stimulus. Many enhancers-promoter-genes with known metabolic function are noted as potential contributors to metabolic disease. It is unclear why Eif6 was chosen for further validation using knockdown, and only in response to TNF α .

Major:

1. It is surprising that there were no changes in the ChIP data given the marked changes in gene expression in response to palmitate and Tnf. It is possible that this is due to unchanged chromatin conformation and interactions, and only differential activation of enhancers and transcription factor binding; however, it is still plausible that the ChIP data are incomplete and/or underpowered. This should be further highlighted in the discussion.
2. The authors began the study by using palm and TNF α to induce IR, but palmitate induced changes are given less emphasis throughout the paper (presumably due to lower magnitude effects on transcription). If the transcriptomic data are overlapped (transcripts altered in both conditions), what ontology terms dominate? Does using the overlapping data (rather than pooled) change the conclusion about enhancer activity and chromatin conformation?
3. The data presented in Figure 1 D-G (showing FDR for top-ranking terms) are not very helpful. Enrichment scores and/or directionality would be more helpful than knowing whether the FDR was 10^{-8} or 10^{-6} . In addition, the rationale for selection of some pathways for demonstrating expression data (Figure 1H-I) is unclear – why were no pathways related to palmitate treatment shown? Why were data related to nucleosome assembly not described further – would be of great interest as that is directly related to stated goals of paper. Why was muscle filament sliding chosen, and not IGF signaling? Can information be provided about the genes responsible for enrichment of these top-ranking ontologies?
4. In the analysis of relationship between K27ac activity and gene expression (lines 170-175), the authors note that promoters connected to enhancers with decreased activity have significantly lower log FC values, and so on for the converse. As currently presented, the text and accompanying Figure 3 do not allow the readers to determine whether there is actually a decrease (negative log FC) or just a lower magnitude increase. Can the data be presented in a way which shows these relationships more clearly?
5. If the authors simply focus on results of palmitate and TNF α incubation (and avoid subsetting by GWAS loci), are the results more concordant with the acute gene expression? I wonder if the GWAS analysis (which was a long-shot to identify genes related to whole-body metabolism in skeletal muscle cells) actually reduces the likelihood of finding relationships between the stimulus and chromatin structure. If the GWAS SNP are actually related to physiology in different cell types (even in muscle itself) or tissues not queried by the myotube analysis, this may actually detract

from the muscle analysis.

6. The knockdown experiments for Eif6 show an impact on OCR, without change in insulin action – findings concordant with association with exercise capacity in mice. Again, however, relating this to body fat distribution GWAS association cannot be justified and detracts from the salient findings. Can expression data for Eif6 be shown in response to palmitate and Tnfa (magnitude, direction), in order to more clearly focus the reader's attention to why this gene is of particular interest for validation?

7. The title indicates that the results identify novel genes controlling whole-body metabolism in humans. I think this should be changed as most of the experiments are performed in cells, with additional analysis of correlation between expression of genes and mouse whole-body phenotypes, and analysis of knockdown in cells. This is not demonstration of control of whole-body metabolism in humans – so the title needs to be revised to tone down the overall conclusion.

8. In the discussion please add more about the differences between palmitate and Tnfa treatment in modulating cellular metabolism and implications for results.

Minor:

1. Of the SNPs identified as related to the putative enhancer function, please provide information about which/how many of these were related to SNPs directly, and how many were related to the SNPs in high LD to the primary SNPs?

2. The description for the x axis value for Figure 1A remains unclear.

3. For Figure 3B and C, the data are not normally distributed- would median be a better metric for describing

4. Line 325 – add "in mice" after GTT.

5. Figure 5 - labels don't match figure labels in legend.

6. Figure S2 – mention panel B in legend.

7. Figure S8 – did protein content change between siScr and siEif6? Did OCR change in response to palmitate or Tnf? This would help to tie back to the original experiment.

Reviewer #1 (Remarks to the Author):

The authors have satisfactorily addressed all my comments.

Kelly A Frazer

Response: We are pleased to hear that the reviewer is satisfied by our revision. The comments were helpful and allowed us to improve the quality of our study.

Reviewer #3 (Remarks to the Author):

In this revised manuscript, Williams and colleagues report results of detailed epigenetic analyses of human myotubes treated with palmitate or TNF α . Integrated analysis of RNA-seq, ChIP seq, and Hi-C together with GWAS data for insulin resistance-related traits results in identification of multiple loci potentially linked to metabolic disease and muscle responses. These data are then used to query mouse tissue expression data to identify genes also correlated with key metabolic traits in mice.

The manuscript is well written and presented clearly. The authors have been responsive to the prior review, adding analysis of mitochondrial oxidative metabolism in cells with knockdown of Eif6. Overall enthusiasm is reduced by the authors' approach to overlap with GWAS, as this may reduce the ability to conclude about transcriptional regulatory effects imposed by the acute stimulus. Many enhancers-promoter-genes with known metabolic function are noted as potential contributors to metabolic disease. It is unclear why Eif6 was chosen for further validation using knockdown, and only in response to TNF α .

Response: We thank the reviewer for the compliments on our study. We have addressed specifically each of the points raised by the reviewer below.

Major:

1. It is surprising that there were no changes in the cHiC data given the marked changes in gene expression in response to palmitate and Tnf. It is possible that this is due to unchanged chromatin conformation and interactions, and only differential activation of enhancers and transcription factor binding; however, it is still plausible that the cHiC data are incomplete and/or underpowered. This should be further highlighted in the discussion.

Response: Thank you for these comments. We would like to point out that reviewers 1 and 2 raised this point in their initial comments. We have provided the following response to reviewers 1 and 2:

“We agree with the reviewer that we cannot rule out that our observation of stable chromatin conformation under palmitate or TNF α treatment was not due to a lack of power in our analysis. Thus, we have re-written this paragraph. However, as mentioned above, our data aligns with a previous study (Jin et al., A high-resolution map of the three-dimensional chromatin interactome in human cells. Nature 2013) where dynamics of promoter–enhancer contacts after TNF α signaling in human fibroblasts show that TNF α -responsive enhancers are already in contact with their target promoters - despite a massive induction of the H3K27ac mark at TNF α -responsive enhancers and in gene transcription.

Another recent paper, detecting dynamic promoter-enhancer contacts during epidermal differentiation, also finds that the majority of differentiation-induced genes have pre-established chromatin contacts - again suggesting that the majority of gene activation is not associated with chromatin remodeling (Rubin et al., Lineage-specific dynamic and pre-established enhancer–promoter contacts cooperate in terminal differentiation. Nature genetics 2017).”

Nevertheless, we now further clarify on this point and also highlight the possibility that our data could be underpowered in discussion page 13.

2. The authors began the study by using palm and TNFa to induce IR, but palmitate induced changes are given less emphasis throughout the paper (presumably due to lower magnitude effects on transcription). If the transcriptomic data are overlapped (transcripts altered in both conditions), what ontology terms dominate? Does using the overlapping data (rather than pooled) change the conclusion about enhancer activity and chromatin conformation?

Response: We have now performed gene ontology analysis of the 1674 transcripts that are altered by both conditions. The results are available to the reviewer at the “GO analysis overlapping transcripts.xlsx” file. In the top 10 gene ontology terms, we did not detect terms common to the gene ontology analyses performed on gene changed after palmitate or TNFa separately. This suggests that the power of gene ontology analysis is decreased by sub-setting with a shorter list of genes. For this reason, we did not opt to include this analysis to our manuscript. We have added the list of genes altered by both conditions in a revised Table S1.

We would like to clarify that the gene expression data was not used in the analysis of enhancer activity or chromatin structure, so the conclusions from the analyses on enhancer activity and chromatin structure would not change if only overlapping transcripts were taken into consideration. In addition, we would like to point out that if we were to use exclusively the overlapping transcripts using our pipeline (illustrated in Figure 4B), we would find only three putative GWAS-target genes that are altered by *both* palmitate and TNFa treatment. This approach would dramatically reduce the number of target genes that we identify and reduce the impact of our study. For this reason, we opted to keep the gene expression analysis separated for palmitate and TNFa.

3. The data presented in Figure 1 D-G (showing FDR for top-ranking terms) are not very helpful. Enrichment scores and/or directionality would be more helpful than knowing whether the FDR was 10^{-8} or 10^{-6} . In addition, the rationale for selection of some pathways for demonstrating expression data (Figure 1H-I) is unclear – why were no pathways related to palmitate treatment shown? Why were data related to nucleosome assembly not described further – would be of great interest as that is directly related to stated goals of paper. Why was muscle filament sliding chosen, and not IGF signaling? Can information be provided about the genes responsible for enrichment of these top-ranking ontologies?

Response: We appreciate this comment and agree that GO analysis results can be more optimally presented. First, we would like to clarify that the GO analysis of differentially expressed genes identified 1,074 and 611 enriched terms for palmitate and TNFa treatment, respectively (FDR<0.01). All these terms, their directionality, and FDR-values are listed in Table S2. In Figure 1 panel D-G, we show, in an unbiased fashion, the top 10 GO terms that are either up- or down-regulated by the respective treatments. To address the reviewer’s comments, we have updated these plots which now visualize the fraction of genes that are differentially expressed along with ontology size and P-values.

We would like to clarify that in panels H and I, we chose to show expression data for the GO-terms “Positive regulation of inflammatory response” as well as “Muscle filament sliding”, as these terms were regulated by both treatments (IGF1 signaling was only regulated by TNFa). Moreover, increased muscle inflammation and lower muscle contraction are both pathways related to skeletal

muscle insulin resistance and therefore we think these pathways were particularly relevant to highlight in our study. We have explained this rationale better in the text in the results section page 5.

4. In the analysis of relationship between K27ac activity and gene expression (lines 170-175), the authors note that promoters connected to enhancers with decreased activity have significantly lower log FC values, and so on for the converse. As currently presented, the text and accompanying Figure 3 do not allow the readers to determine whether there is actually a decrease (negative log FC) or just a lower magnitude increase. Can the data be presented in a way which shows these relationships more clearly?

Response: The reviewer is right that the data visualization that we opted for only informs about relative, and not absolute, fold changes. This representation was in fact intentional, as it allows to illustrate the main point that enhancer activity is related to gene transcription. To the best of our efforts, we could not think of a better way to illustrate these results.

5. If the authors simply focus on results of palmitate and TNF α incubation (and avoid subsetting by GWAS loci), are the results more concordant with the acute gene expression? I wonder if the GWAS analysis (which was a long-shot to identify genes related to whole-body metabolism in skeletal muscle cells) actually reduces the likelihood of finding relationships between the stimulus and chromatin structure. If the GWAS SNP are actually related to physiology in different cell types (even in muscle itself) or tissues not queried by the myotube analysis, this may actually detract from the muscle analysis.

Response: We would like to clarify that we did not subset the data based on GWAS information before analyzing the association between chromatin conformation and gene expression changes. Moreover, we would like to reiterate that our GWAS-based analysis constitutes a substantial added value of our study, and allows to get insight into muscle-driven whole-body phenotype in humans, which cell and animal model cannot provide. For this reason, we strongly believe that overlapping with GWAS data should be kept in our study.

6. The knockdown experiments for Eif6 show an impact on OCR, without change in insulin action – findings concordant with association with exercise capacity in mice. Again, however, relating this to body fat distribution GWAS association cannot be justified and detracts from the salient findings. Can expression data for Eif6 be shown in response to palmitate and Tnfa (magnitude, direction), in order to more clearly focus the reader's attention to why this gene is of particular interest for validation?

Response: The reviewer raises a valid point but in fact, expression data for *EIF6* after palmitate and TNF α treatment was already shown in Figure 6C. Yet, we realize that Figure 6B and C are missing legends appropriately describing color-codes within the figures. We apologize for this mistake and we have corrected Figure 6B and C so it should be clear now.

We think that our investigations of Eif6 do not distract from the main message of our study. In fact, skeletal muscle mitochondrial respiration is linked to WHR (*Bharadwaj et al., Relationships between mitochondrial content and bioenergetics with obesity, body composition and fat distribution in healthy older adults. BMC Obes. 2015*) and variants in mitochondrial genes are associated with WHR (*Kraja et al., Associations of Mitochondrial and Nuclear Mitochondrial*

Variants and Genes with Seven Metabolic Traits. Am J Hum Genet 2019). To make this even more clear, we have included the latter reference in the discussion page 16.

7. The title indicates that the results identify novel genes controlling whole-body metabolism in humans. I think this should be changed as most of the experiments are performed in cells, with additional analysis of correlation between expression of genes and mouse whole-body phenotypes, and analysis of knockdown in cells. This is not demonstration of control of whole-body metabolism in humans – so the title needs to be revised to tone down the overall conclusion.

Response: The reviewer raises a fair point. We have changed the title from: "Skeletal muscle enhancer interactions identify novel genes controlling whole body metabolism in humans" to "Skeletal muscle enhancer interactions identify novel genes controlling whole-body metabolism".

8. In the discussion please add more about the differences between palmitate and Tnfa treatment in modulating cellular metabolism and implications for results.

Response: We have included a section discussing the effect of palmitate and TNFa treatment on cellular metabolism and insulin resistance in the discussion section page 13-14.

Minor:

1. Of the SNPs identified as related to the putative enhancer function, please provide information about which/how many of these were related to SNPs directly, and how many were related to the SNPs in high LD to the primary SNPs?

Response: Thank you for this comment. We detected 390 disease-associated SNPs within the 11 palmitate- and 124 TNFa-regulated enhancers depicted in Figure 4B. Out of these, we find 15 to be primary GWAS SNPs and 375 to be SNPs in high LD to a primary SNP. We have now included this information in Table S5, so that primary, *versus* linked SNPs can be identified.

2. The description for the x axis value for Figure 1A remains unclear.

Response: We have clarified in Figure 1 legend that “dim” means “dimension”.

3. For Figure 3B and C, the data are not normally distributed- would median be a better metric for describing

Response: We agree with the reviewer, and we have now included the median for describing the data instead of the mean

4. Line 325 – add “in mice” after GTT.

Response: Thank you. We have added this.

5. Figure 5 - labels don't match figure labels in legend.

Response: We apologize for this mistake. It has been corrected.

6. Figure S2 – mention panel B in legend.

Response: Thank you. This has been added.

7. Figure S8 – did protein content change between siScr and siEif6? Did OCR change in response to palmitate or Tnf? This would help to tie back to the original experiment.

Response: We did not measure protein levels of EIF6. While this could be an addition to our study, we think the main finding that *Eif6* expression alter mitochondrial respiration infers that EIF6 protein plays a role.

We would like to point out that previous studies have shown that TNF α treatment of C2C12 cells lowers OCR (McLean et al., Tumor necrosis factor- α (TNF) effects on mitochondrial metabolism in C2C12 myotubes. FASEB Journal 2013). Thus, our findings that TNF α treatment lowers *Eif6* expression, and that *Eif6* knockdown lowers OCR is consistent with the literature.